# Unique Habitat of Karst Tiankengs Changes the Taxonomy and Potential Metabolism of Soil Microbial Communities

Cong Jiang,[a] ⬤Hui Zeng[a]

[a]School of Urban Planning and Design, Peking University Shenzhen Graduate School, Peking University, Shenzhen, China

**ABSTRACT**   Microbial communities in karst ecosystems have been extensively studied. However, in a class of deep-lying habitats with unique climates (karst tiankeng), the structure and ecological functions of microorganisms receive little attention, which is essential for understanding the biogeochemistry of karst tiankeng. Herein, microorganisms from inside (ITK) and outside (OTK) karst tiankengs were analyzed by high-throughput sequencing and multivariate statistical analysis. The results showed that the structure and function of soil bacterial communities inside and outside karst tiankengs were significantly different. The ITK microbial communities presented significantly higher Shannon diversity due to the abundant nutrients in karst tiankeng soil. Random molecular ecological network analysis revealed that the ITK network was simpler and more vulnerable and may be susceptible to environmental changes. More positive links within the network indicate that microorganisms adapt to the karst tiankeng through synergies. The keystones in karst tiankeng were mainly involved in the decomposition of soil organic matter and carbon/nitrogen cycles. Although soil total phosphorus and available potassium regulate microbial community structure variation, dispersal limitation is the predominant ecological process within the microbial community in karst tiankeng. In addition, the functional profiles of the microbial communities reveal that some human diseases (such as infectious diseases) exist in OTK. Collectively, these findings have enhanced our understanding of microbial interactions, ecological functions, and community composition processes in karst tiankeng ecosystems.

**IMPORTANCE**   Constrained by the trapped terrain, a unique ecosystem has formed in karst tiankeng. Soil microorganisms are essential for the formation and maintenance of ecosystems, but soil microbial ecology research in karst tiankeng is still lacking. In this study, representative habitats inside and outside karst tiankeng were selected to study the taxonomy and potential metabolism of soil microbial communities. The results show that the unique habitat of karst tiankeng reshapes the composition, structure, and function of soil microbial communities. Our results contribute to enhancing our understanding of sustainable recovery strategies in fragile ecosystems and understanding the biodiversity value of karst tiankeng under climate change.

**KEYWORDS**   community assembly, cooccurrence network, functional prediction, karst tiankeng, soil microbial community

The world's largest karst region is located in China, accounting for approximately 1/3 of the country's land area (1). In the past 2 decades, a large-scale negative topography karst landscape has been discovered in the karst area, which grows under the specific conditions of lava geology, climate, and hydrological environment in southern China (2). Zhu and Waltham (3) defined extraordinary spatial and morphological characteristics such as large volumes, steep and enclosed rock walls, deep well-like or barrel-like contours, plane widths and depths ranging from more than 100 m, and a series of super-large karst negative terrain with the bottom connected to the underground river as a "tiankeng." The isolation effect of the vertical cliffs makes the internal habitat of the

**Editor** Frédérique Reverchon

Address correspondence to Hui Zeng, zengh@pkusz.edu.cn.

The authors declare no conflict of interest.

karst tiankeng independent of the external environment, and a unique primitive microclimate with low temperatures, high humidity, and low solar radiation has been formed (4, 5). The karst tiankeng ecosystem breeds rich and unique biological resources. The karst tiankeng flora is characterized by high diversity and strong originality and preserves ancient and unique plant species such as alder and cool-adapted plants. The karst tiankeng is also a paradise for rare animals, e.g., *Belisana zhangi* sp. nov., *Prionodon pardicolor*, and *Sinocyclocheilus hyalinus*. Under global climate change, karst tiankengs could become biodiversity conservation reservoirs and sanctuaries for some important species (6, 7).

Highly diverse microbial communities are an important part of soil biological systems and play a key role in maintaining ecosystem functions, including degradation and humification of organic compounds in soil ecosystems (8, 9). Habitat heterogeneity significantly affects changes in biodiversity patterns and alters soil chemistry, vegetation communities, and microbial communities (10, 11). More heterogeneous habitats generally generate increased variation in microbial community composition and can support higher species alpha diversity (12, 13). Correspondingly, microbes adopt different survival strategies in different habitats (14).

Karst areas are climate change-sensitive areas and are $CO_2$ sinks that cannot be ignored (15, 16). In view of the vital regulatory role of microorganisms in regional biogeochemical cycles, the characteristics of microbial communities in karst regions have been studied (17–19). In fact, mineral substrate significantly impacts bacterial and fungal communities in karst caves (20). Significant differences in microbial community diversity have also been found inside (ITK) and outside (OTK) the karst caves (21). The results of Yun et al. (22) indicated that compositional variability among microbial communities in the different karst cave habitats reflect spatial pH changes. In the isolated karst tiankeng ecosystem, microbial living activities and interactions are essential for maintaining community stability and ecosystem function. However, our understanding of the structure, function, and assembly processes of the microbial community in karst tiankeng remains very limited.

Due to the differences in microclimate, soil nutrients, and vegetation cover from inside and outside of the karst tiankeng, we hypothesize that (i) the unique habitat of karst tiankeng alters structure and function of soil microbial communities, and (ii) karst tiankeng maintains a complex and stable soil microbial network. To test these hypotheses, we studied the soil bacterial communities from inside and outside of the karst tiankeng. By using high-throughput sequencing of the 16S rRNA gene, we aimed to (i) assess the taxonomic and functional diversity of microbial communities from inside and outside of the karst tiankeng and (ii) determine the assembly processes and molecular ecological network of microbial communities in karst tiankeng.

## RESULTS

**Structures of bacterial communities in karst tiankeng soils.** The rarefaction curves of all soil samples were saturated (see Fig. S1 in the supplemental material) and indicated that the sequencing depth was sufficient to cover most microorganisms. A total of 2,247,020 quality sequences were obtained from all soil samples and were grouped into 16,630 amplicon sequence variants (ASVs) (ranging from 1,308 to 1,807 ASVs). Among the ASVs, 39.31% (6,438 ASVs) and 39.15% (6,511 ASVs) were uniquely observed in the ITK and OTK sites, respectively, and 21.54% (3,581 ASVs) were shared by the ITK and OTK soil samples (see Fig. S2 in the supplemental material).

The Shannon index was significantly higher at the ITK sites than at the OTK sites ($P < 0.05$), while the Chao1 index was not significantly different between the ITK and OTK sites ($P > 0.05$) (see Fig. S3 in the supplemental material). The principal coordinate analysis (PCoA) indicated that bacterial communities in the ITK and OTK sites had obvious differentiation effects (Fig. 1). Furthermore, the permutational multivariate analysis of variance (PERMANOVA), analysis of similarity (ANOSIM), and multiresponse permutation procedure (MRPP) analysis results showed that bacterial community

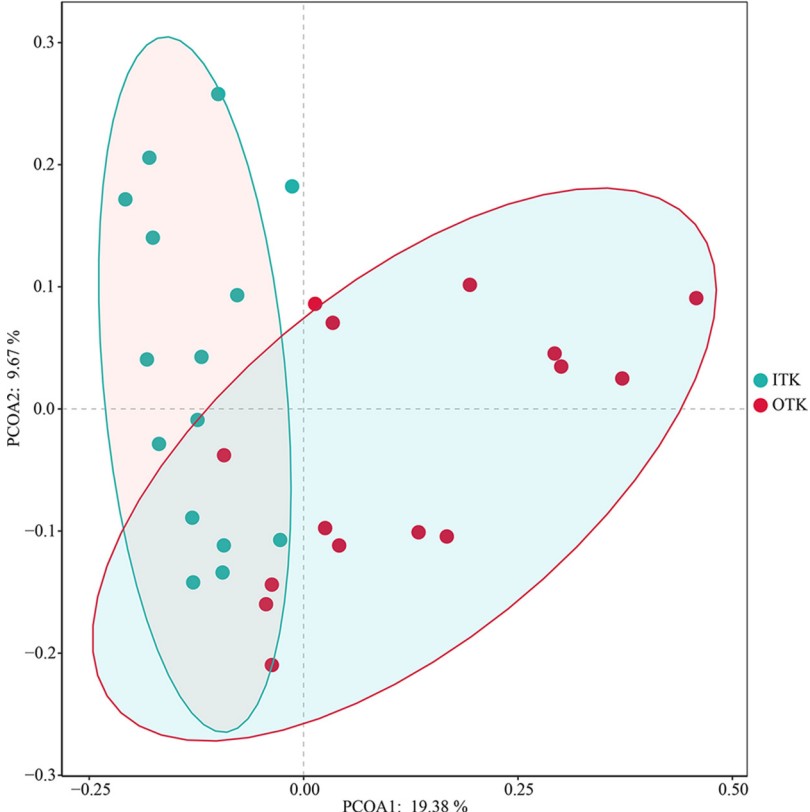

**FIG 1** The principal coordinate analysis (PCoA) of microbial community composition inside and outside karst tiankeng. ITK, inside the tiankeng; OTK, outside of the tiankeng.

dissimilarity was significantly different between the ITK and OTK soils (see Table S1 in the supplemental material). The bacterial community composition at the phylum level (top 10) is shown in Fig. S2. *Proteobacteria* was the most abundant phylum, and the average abundance were 40.76% and 43.18% in ITK and OTK, respectively. The second dominant phylum was *Actinobacteria* with average abundances of 20.77% and 16.14% in ITK and OTK, respectively. Significant differences in the mean proportion of classes were shown in Fig. S4 in the supplemental material; *Thermolephilia*, *Acidobacteria_6*, and *Actinobacteria* were more abundant in ITK sites, whereas the *Alphaproteobacteria*, *Acidobacteriia*, and *DA052* were more abundant in OTK sites. Furthermore, linear discriminant analysis effect size (LEfSe) analysis results showed that the 70 microbial clades exhibited significant differences between the ITK and OTK sites (linear discriminant analysis [LDA] score, >3.0) (Fig. 2). Most differential bacteria were significantly enriched in ITK (51). More specifically, *Actinobacteria* (phyla), *Gammaproteobacteria* (class), *Actinomycetales* (order), and *Pseudomonas* (genus) were abundant in ITK; *Burkholderia* (genus), *Burkholderiaceae* (family), and *Bradyrhizobium* (genus) were abundant in OTK.

**Quantifying community assembly processes in karst tiankeng soils.** Mantel correlograms consistently indicated significant correlations across short phylogenetic distances in both ITK and OTK soils ($P < 0.05$) (see Fig. S5 in the supplemental material). Therefore, the phylogenetic turnover rate between recent kinships (e.g., nearest taxon index [NTI] and $\beta$-nearest taxon index [$\beta$NTI]) is an ecological corollary suitable for this study. The NTI results also suggested that the soil microbial communities in karst tiankeng were clustered phylogenetically rather than by chance (see Table S2 in the supplemental material). The $\beta$NTI values of the ITK and OTK sites were between $-2$ and $+2$, indicating that stochastic processes controlled microbial community assembly (Fig. 3A). The ecological processes affecting microbial community construction were

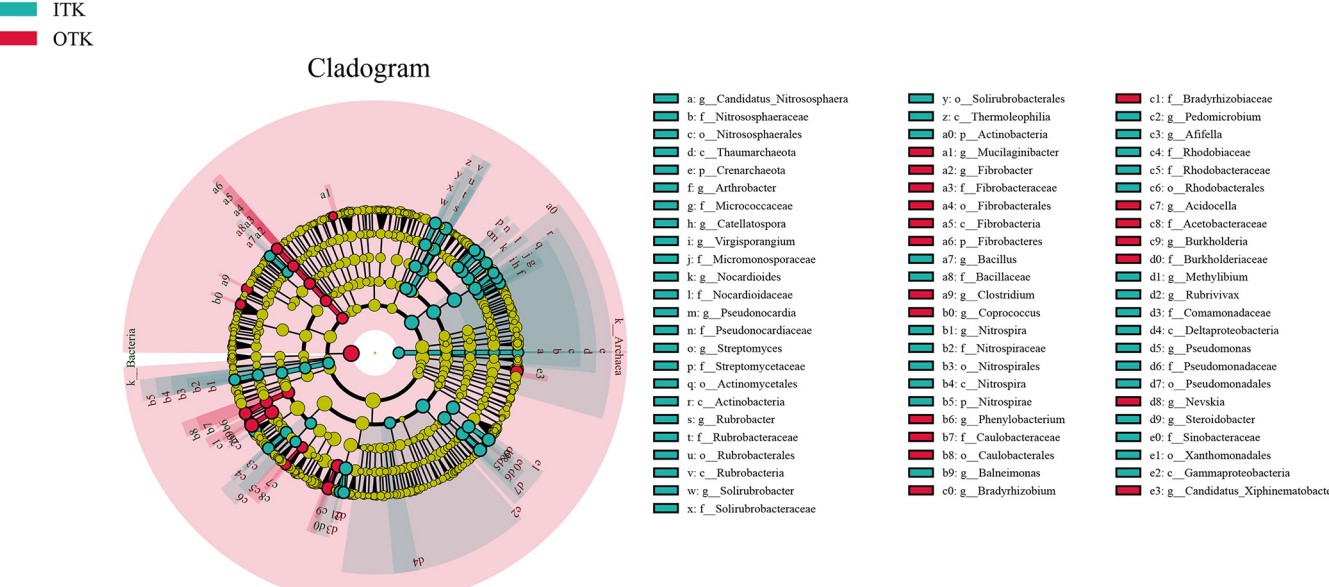

**FIG 2** The LEfSe analysis of microbial communities inside and outside karst tiankeng.

further refined, and the results showed that dispersal limitation was the most important process affecting the aggregation of microbial communities, followed by homogeneous selection (Fig. 3B).

**Relationships between microbial community and soil properties.** We found that the soil physiochemical properties varied between the inside and outside the tiankeng sites. Soil samples from ITK showed significantly higher soil water content (SWC), total nitrogen (TN), available potassium (AK), available phosphorus (AP), and available nitrogen (AN) than those from OTK sites ($P < 0.05$) (Table 1). ITK soils had higher contents of total organic carbon (TOC), total phosphorus (TP), and total potassium (TK), and no significant differences were detected between ITK and OTK sites. The soil samples were all slightly acidic, with pH values ranging from 6.27 to 6.30 (Table 1). The tiankeng hab-

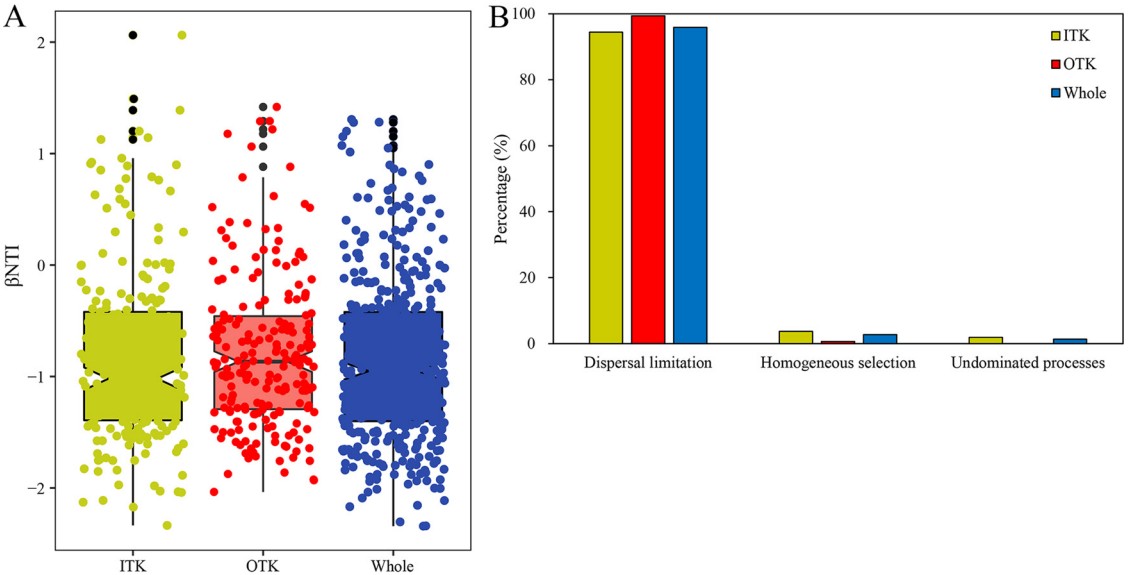

**FIG 3** (A) The assembly processes of microbial community in inside and outside karst tiankeng. (B) The percentage of each ecological processes in stochastic processes. Whole, all karst tiankeng sites of bacterial community.

**TABLE 1** Physiochemical properties of soil inside and outside of the karst tiankeng[a]

| Location | SWC | TOC (g/kg) | TN (g/kg) | TP (g/kg) | TK (g/kg) | AK (mg/kg) | AP (mg/kg) | AN (mg/kg) | pH |
|---|---|---|---|---|---|---|---|---|---|
| ITK | 0.33 ± 0.10a | 51.59 ± 27.86a | 3.87 ± 1.85a | 716.64 ± 214.65a | 4.69 ± 1.47a | 179.80 ± 51.40a | 1.03 ± 0.05a | 275.52 ± 105.70a | 6.27 ± 0.31a |
| OTK | 0.26 ± 0.09b | 31.75 ± 18.67a | 1.90 ± 1.17b | 407.73 ± 178.00a | 3.56 ± 2.02a | 133.87 ± 57.40b | 1.00 ± 0.03b | 177.73 ± 100.74b | 6.30 ± 0.32a |

[a]ITK, inside tiankeng; OTK, outside tiankeng; SWC, soil water content; TOC, total organic carbon; TN, total nitrogen; TP, total phosphorus; TK, total potassium; AK, available potassium; AP, available phosphorus; AN, available nitrogen. Values are means ± standard error (SE). Values within the same row not followed by the same letter differ significantly ($P < 0.05$).

itat also changed the plant characteristics. The Shannon-Wiener index and species richness were higher at the ITK sites, while the difference with OTK sites was not significant ($P > 0.05$) (see Fig. S6 in the supplemental material).

The taxonomic diversity was correlated with TN ($r = 0.45$; $P < 0.05$), TP ($r = 0.48$; $P < 0.05$), AK ($r = 0.40$; $P < 0.05$), and AN ($r = 0.38$; $P < 0.05$) (see Table S3 in the supplemental material). The major bacterial phyla were also correlated with the soil physiochemical properties (see Fig. S7 in the supplemental material). Redundancy analysis (RDA) axes 1 and 2 (RDA1 and RDA2) explained 25.45% and 18.99% of the variation in the microbial community, respectively (Fig. 4A). The RDA results showed that TP and AK were the key soil variables that determined the composition of the bacterial communities. In addition, TP and AK in the partial Mantel test showed significant relationships with the microbial community composition in karst tiankeng ($P < 0.05$) (Fig. 4B).

**Bacterial networks and keystone taxa in karst tiankeng soils.** Molecular ecological networks (MENs) can reflect potential microbial-microbial interactions in karst tiankeng soils. Both ITK and OTK molecular ecological networks exhibit the characteristics of scale-free networks ($R^2$ values were 0.907 and 0.912, respectively) (Table 2). In total, 381 nodes and 705 links were shared in ITK networks; 352 nodes and 863 links were shared in OTK networks. Both modularity values were greater than 0.4, indicating that the constructed networks had a good modular structure of microbial communities. The network of OTK was more complex than that of ITK, as indicated by the higher values of average degree (avgK) and shorter average path distance (GD). Higher values of the average clustering coefficient (avgCC) and density (D) were observed in ITK (Table 2). More positive interaction edges (74.58%) were founded in the ITK network than in the OTK network (72.01%) (Table 2). Furthermore, we analyzed the phylogenetic composition of modules (>5% of the total number of modules) in each network (Fig. 5). The module composition generally differed between inside and outside the tiankeng habitats (ITK and OTK soils). In the network of ITK soil, six modules were observed. Modules 0, 1, 2, and 5 were dominated by *Proteobacteria* taxa; *Proteobacteria* and *Actinobacteria* co-occurred relatively evenly in module 4; *Actinobacteria* was predominant in module 3 (Fig. 5A). In the OTK soil network, module 0 was the largest module. *Proteobacteria*, *Acidobacteria*, and *Actinobacteria* dominated in most modules.

Natural connectivity analysis was conducted to determine the robustness of the network in bacterial communities of ITK and OTK soils. As shown in Fig. S8A in the supplemental material, the robustness value of the microbial networks in OTK was higher than that in ITK. The results indicated that karst tiankeng habitats had an important impact on the robustness of soil microbial networks. Furthermore, robustness and vulnerability indices were used to characterize the stability of the network structure (Fig. S8B and C). In the patterns of random species loss and targeted removal of models, the robustness value of the OTK network was higher than that of the ITK network. The network vulnerability was also lower in OTK than in ITK.

The core nodes of the network can be identified by the within module connectivity ($Zi$) and the among-module connectivity ($Pi$) (Fig. 6). According to the results, 94.75% and 95.74% were peripherals in the ITK and OTK soils, respectively. A total of 19 keystone taxa were observed in the network of ITK soils. Nine keystone taxa were module hubs, and the rest were connectors. In contrast, 11 keystone taxa (7 module hubs and 4 connectors) were identified in the ITK network (see Table S4 in the supplemental material). *Proteobacteria* taxa predominated in all keystone taxa, which accounted for 57.90% and 45.46% of all keystone taxa in the ITK and OTK networks, respectively. *Actinobacteria*, *Acidobacteria*,

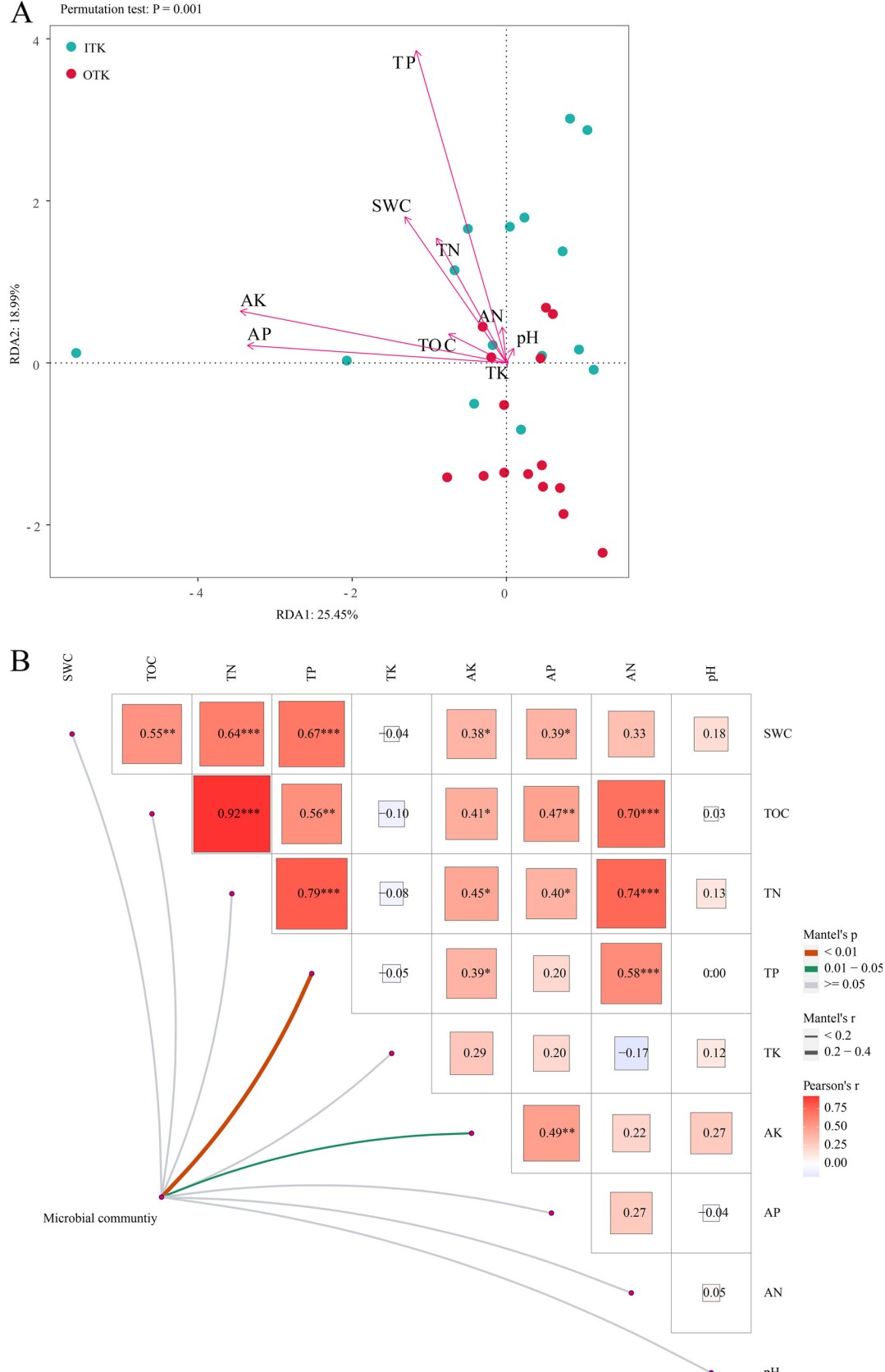

**FIG 4** Redundancy analysis (RDA) (A) and partial Mantel test (B) of soil properties and microbial communities inside and outside of the karst tiankeng. SWC, soil water content; TOC, total organic carbon; TN, total nitrogen; TP, total phosphorus; TK, total potassium; AK, available potassium; AP, available phosphorus; AN, available nitrogen. Values within the same row not followed by the same letter differ significantly ($P < 0.05$).

**TABLE 2** Key topological features of inside and outside karst tiankeng soil microbial networks[a]

| Location | No. of nodes | No. of Edges | $R^2$ of power law | avgCC | Avg path | Avg degree | GD | CD | Density | Modularity | Positive interaction edges (%) |
|---|---|---|---|---|---|---|---|---|---|---|---|
| ITK | 381 | 705 | 0.912 | 0.114 | 4.613 | 3.701 | 4.995 | 0.083 | 0.010 | 0.688 | 74.58 |
| OTK | 352 | 863 | 0.907 | 0.132 | 4.995 | 4.847 | 4.613 | 0.101 | 0.014 | 0.572 | 72.01 |

[a]avgCC, average clustering coefficient; GD, average path distance; CD, centralization of degree.

*Bacteroidetes*, *Verrucomicrobia*, *Firmicutes*, and *Gemmatimonadetes* were the keystone taxa in the ITK network. In contrast, *Verrucomicrobia*, *Planctomycetes*, *Actinobacteria*, and *Acidobacteria* were the keystone taxa in the OTK network. Furthermore, more keystone taxa belonged to modules 2 and 3 in the ITK network; more keystone taxa belonged to module 1 in the OTK network (Table S4).

**Microbial functional prediction in karst tiankeng soils.** The potential functions of karst tiankeng soil bacterial communities were annotated, and a total of 7,342 KEGG orthology (KO) genes were screened. Metabolism predominated in all potential functions, which accounted for 74.54% and 74.31% of all potential functions in ITK and OTK, respectively (see Fig. S9 in the supplemental material). The potential functional profiles (KEGG level 2) of ITK and OTK soil were compared by STAMP, and it was found that the functional gene categories associated with amino acid metabolism, metabolism of terpenoids and polyketides, global and overview maps, nervous system, and substance dependence were significantly enriched in the ITK soil. Functional genes associated with chemical structure transformation maps, cell motility, glycan biosynthesis and metabolism, cell growth and death, and membrane transport were more abundant in OTK. In addition, we found that bacteria communities related to human diseases (infectious disease, bacterial and viral) were significantly higher in OTK (Fig. 7). Based on the KEGG database, the genes related to the C and N cycles were identified (see Fig. S10 in the supplemental material). Most of the C cycle showed higher abundances in the outside karst tiankeng (OTK). The abundances of the genes associated with nitrogen metabolism, nitrogen fixation, and ammonification were also higher in OTK.

## DISCUSSION

**Microbially distinct taxonomic and functional diversity in karst tiankeng soils.** In our study, diverse taxonomic and functional bacterial communities were detected inside

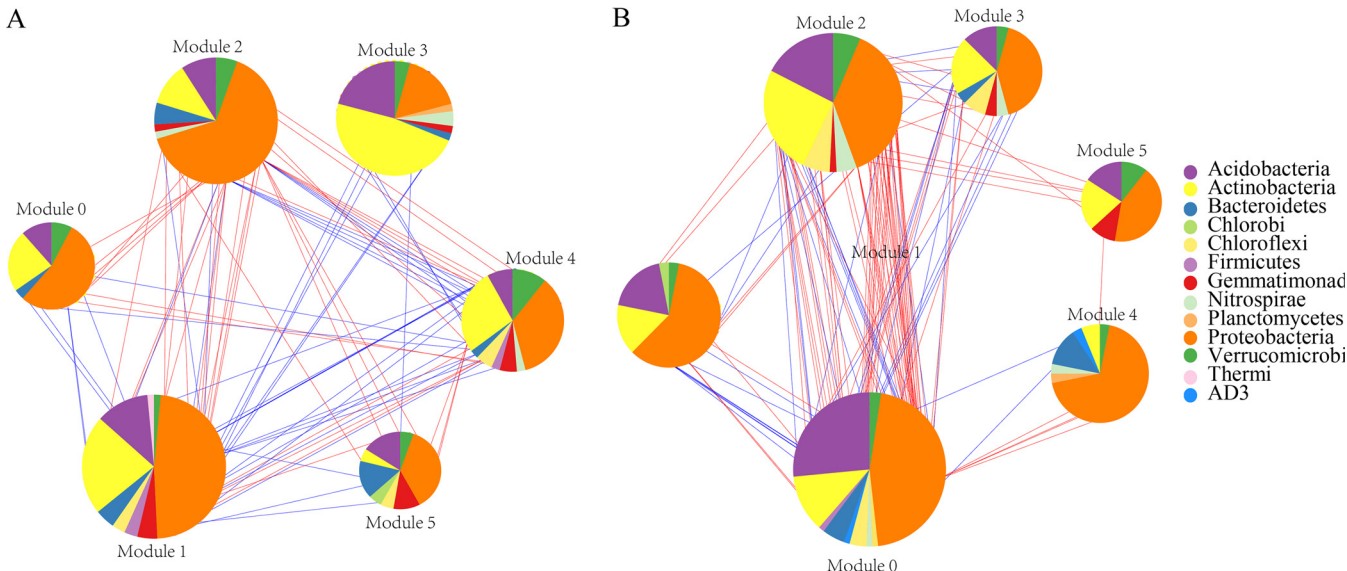

**FIG 5** Phylum-level composition of the dominant modules inside (A) and outside (B) karst tiankeng networks.

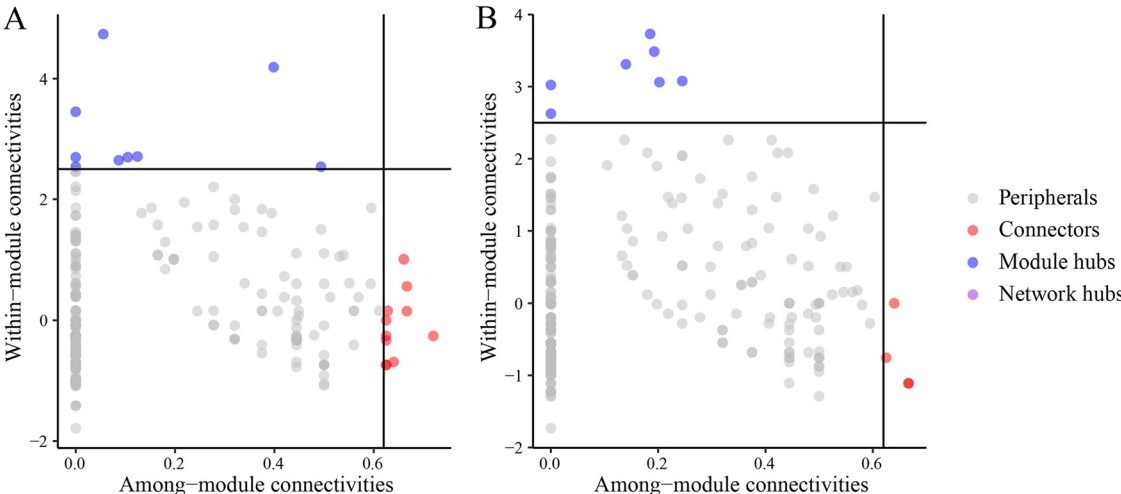

**FIG 6** Classification of nodes to identify keystone species within the networks of inside (A) and outside (B) of the karst tiankeng.

and outside karst tiankeng soils with a dominance of *Proteobacteria* and *Actinobacteria*, which is consistent with previous study of karst soils (23–26). *Proteobacteria* are considered to have strong oxidation capacity of organic and inorganic compounds and harvest energy from light (27). The dominant phylum of *Proteobacteria* in karst tiankeng may suggest that microbes that perform the matter decomposition and nutrient cycles survive well in karst tiankeng. The same microbial taxa with different relative abundances were observed inside and outside of the tiankeng sites. For example, *Actinobacteria* are fast-growing copiotrophs and favor organic matter-rich soils (28). The relative abundance of *Actinobacteria* was higher in ITK soils (see Fig. S2 in the supplemental material), which might be because the soil in the tiankeng was more nutritious. This result is also confirmed by the significantly higher soil nutrient contents of TN, AK, AP, and AN in ITK (Table 1). Compared with ITK soils, higher relative abundance of *Acidobacteria* and *Bacteroidetes* was observed in OTK soils because they survive well in nutrient-poor soil environments (29). Outside of the tiankeng are typical fragile karst ecosystems characterized by water scarcity and soil erosion (30). The thin soil layer combined with the developed karst system makes soil outside of the tiankeng nutrient scarce, which can be supported by the results of the soil physiochemical properties (Table 1). Furthermore, LEfSe analysis indicated that inside

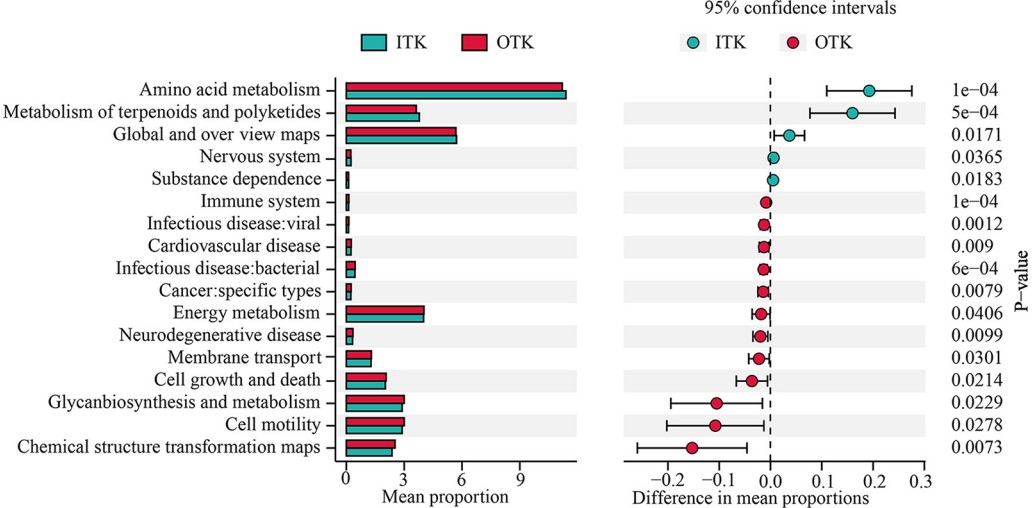

**FIG 7** STAMP analysis of functional gene categories (KEGG level 2) inside and outside of the karst tiankeng ($P < 0.05$).

and outside karst tiankeng soils harbored different biomarkers. These results support our first hypothesis. Most biomarker bacteria in ITK belonged to *Actinobacteria*, possibly resulting from their ability to decompose organic matter. Rich vegetation in ITK sites tends to represent more litter available for soil microbial decomposition (see Fig. S6 in the supplemental material) (31, 32). The indicator taxa in ITK also included *Micromonosporaceae* and *Nitrospira* (Fig. 2). *Micromonosporaceae* play a key role in enhanced cellulose-degrading capability and possibly favored rich soil organic C condition (33, 34). *Nitrospira* has been illustrated to be sensitive to acid and fertilized soils (35). It should be noted that the karst tiankeng is characterized by fertile soil and rich vegetation, and it was not surprising to observe these biomarkers in ITK. The biomarkers in OTK included *Phenylobacterium* and *Clostridium*. These biomarkers were related to anthropogenic activities, such as *Clostridium*, a sensitive indicator for livestock fecal pollution (36, 37). Due to the limitations of topography, karst tiankeng can be less disturbed by agriculture and grazing activities. Overall, these results strongly support the vital role played by unique habitats of karst tiankeng in influencing microbial taxonomic and functional diversity.

Understanding functional group responses to karst tiankeng unique habitats is critical to predicting karst tiankeng ecological processes. In this study, functional analysis demonstrated the difference in microbial communities between the inside and outside of the karst tiankeng (Fig. 7). Most C cycle-related genes were found to have higher abundances in OTK. In addition, we observed a higher potential for nitrogen metabolism, nitrogen fixation, and ammonification in OTK (see Fig. S9 in the supplemental material). The "nutrient limitation theory" can explain these results well (38). When nutrients are deficient in the soil, increased expression of C and N cycle genes in the microbial community contributes to the breakdown of organic matter, thereby increasing soil nutrients (e.g., C and N). When the soil nutrients are sufficient, the microbial community easily obtains nutrients (Table 1), and the expression of C and N cycle genes will decrease, thus reaching a stable state (39). In addition, we also found that gene sequences related to human diseases (such as neurodegenerative disease and infectious bacterial and viral diseases) were more abundant in OTK (Fig. 7). Zhou et al. (14) showed that alpine grassland degradation increases the risk of bacteria involved in human disease, and use of livestock without examination in this area has caused *Salmonella* infections. The karst tiankeng in this study is located in a typical karst degradation area, which is characterized by severe rocky desertification, sparse vegetation, and a large disturbance of human activities (e.g., grazing). The ecological environment of the OTK is at great risk, and the loss of biodiversity may have a negative health effect. Correlation has been observed between loss in biodiversity and an increase in human disease frequency (40, 41). In contrast, limited by topography, the karst tiankeng is less disturbed by anthropogenic activities, and the higher soil nutrients and vegetation cover conditions are formed, which creates suitable conditions for the survival of microorganisms. Higher bacterial community diversity was observed in ITK (see Fig. S3 in the supplemental material). Sandifer et al. (42) reported that environmental bacterial profiles are closely associated with human diseases, and exposure to microbial biodiversity help to reduce certain human diseases. Human exposure to high soil microbial diversity increases the chance of collisions between microbiomes and thus avoids the loss of specific microbiome traits (43, 44). Although the relationship between karst ecosystems and human health requires further human disease investigations to determine, this phenomenon is noteworthy.

**Characteristics of microbial networks in karst tiankeng soils.** The complexity of the network depends on the interaction type of microbial communities (e.g., antagonistic, competitive, or mutualistic) (45). The inner microbial interactions within the karst tiankeng remain unknown. In karst tiankengs with unique habitats, microorganisms may adopt different strategies to survive. Interactions between microorganisms are determined by the basic dynamics of each species to promote their survival and are critical to community stability (46, 47). The OTK network is more complex than the ITK network, indicating that OTK has a greater potential for interactions of different

microbial groups. Microbial communities resist environmental disturbances by forming strong interspecific interactions (48). The microbial community of OTK adapts to complex environmental changes by forming complex interactions. The average path length of the ITK soil network is smaller than that of the OTK soil (Table 2), which is characteristic of a small world network and enables the karst tiankeng ecosystem to respond quickly to disturbances. Therefore, karst tiankeng microbial communities may be highly sensitive to environmental changes. Both the ITK and OTK soil networks had a high proportion of positive interaction edges, which indicated the importance of synergies between microorganisms in karst tiankeng habitats (49). Microbial collaboration essentially increases the capacity of communities to adapt to karst tiankeng ecosystems. Previous studies have interpreted modules as niches with significant implications for ecosystem stability (50, 51). The modularity values were higher in the ITK soil network than in the OTK soil network, which may be linked to stronger niche differentiation in the ITK soil than in the OTK soil. In our study, niche differentiation leads to a decrease in the stability of microbial networks (Table 2). Main modules are closely related to environmental factors, and habitat heterogeneity might affect microbial networks through main modules (52). The differences in the number of nodes and taxa of the main modules in ITK and OTK networks confirmed that habitat heterogeneity plays an important role in module formation. The OTK soil had the more stable microbial networks in comparison with those of the ITK soil due to high robustness values, which contradicts our second hypothesis (see Fig. S8 in the supplemental material). Several studies have shown that competition within the microbial community is more intense under poor nutritional conditions, and competition between microbes enhances the robustness of the network (53, 54). This phenomenon was also observed in this study with higher negative interaction edges in OTK than in ITK soils (Table 2).

Identifying keystone species in a community is important for communities with a high degree of diversity and complexity (55, 56). Keystones have the effect of stabilizing community structures (57). Compared to complicated OTK networks, more modular hubs and connectors were identified in ITK networks (Table S4), indicating that keystones in karst tiankeng contribute more to maintaining multiple ecological functions. *Pseudonocardia*, *Bacillales*, and *Alphaproteobacteria* were identified as the keystones in ITK soil, which might play key roles in decomposition of various macromolecular organic matter and participate in the global cycling of carbon, nitrogen, and phosphorus (14, 58, 59). *Verrucomicrobia*, *Planctomycetes*, and *Acidobacteria* were identified as the keystones in OTK soil. These microorganisms have participated in the formation of karst soils throughout geological history. Different keystones play an important role in maintaining differentiated community structure and function inside and outside karst tiankeng.

**Relationship between microbial community and soil environmental variables.** Previous studies have indicated that the structure and composition of microbial communities changed as the habitat changed (60, 61). Different habitats are often accompanied by changes in soil condition, vegetation cover, and microclimate (62). Soil environmental variables are one of the most key factors shaping the structure and composition of soil microbial communities (63). Karst tiankeng sinks deep in the surface, and unique habitats form within the tiankeng that are distinct from those outside of the tiankeng (typical of degraded karst terrain). Our study showed that soil properties varied between the inside and outside the karst tiankeng (Table 1), with significant correlations between microbial taxonomic diversity and TP, TN, AK, and AN (see Table S3 in the supplemental material). These results suggest that differences in soil microbial taxonomic diversity within and outside karst tiankeng may be due to soil properties. The RDA results showed that TP and AK were the key drivers of the microbial community in karst tiankeng (Fig. 4A). In addition, the partial Mantel test corroborates this result (Fig. 4B). Liu et al. (64) reported that TP is a limiting factor of microbial communities. Karst areas are generally subject to phosphorus restrictions, and thus TP content affects the soil microbial communities (17). The soil AK content mainly comes from the accumulation of effective nutrients (65). Habitat changes significantly affect AK content, which in turn affects soil microbial communities. The

changes in AK in karst tiankeng soils may be affected by the absorption of vegetation and litter input, which likely affect microbial communities by providing a variety of microbial growth resources (66). Habitat changes drive changes in the microbial community, which may indirectly shape the unique structure of karst tiankeng microbial communities by influencing soil properties such as TP and AK. Although soil properties obviously regulate bacterial community structure variation, our results also showed that in karst tiankeng environments, stochastic processes caused by dispersal limitation still exist within the bacterial community. Our study suggested that the role of stochastic processes in the formation of microbial community diversity cannot be ignored.

**Conclusion.** This study revealed the structure, potential function, and assembly processes of soil microbial communities in karst tiankengs in karst areas. The results demonstrated that karst tiankeng habitat dramatically impacts soil microbial communities in various ways. The microbial Shannon diversity index significantly differed inside and outside of the karst tiankeng, and the microbial community structure was significantly affected by the karst tiankeng. Karst tiankengs have unique biomarkers, keystone taxa in the network, and distinctive functions. The functional pathways of the soil microbial communities outside karst tiankengs were found to be more involved in human diseases, such as infectious diseases. Dispersal limitation is the most important process of microbial community assembly. The microbial communities are mainly related to TP and AK. The microbial network in the karst tiankeng is simple and vulnerable, which may indicate that karst tiankeng microbes are more susceptible to human activities and climate change. These results contribute to the evaluation of the uniqueness of karst ecosystems and provide a more comprehensive understanding of karst tiankeng microbial ecology.

## MATERIALS AND METHODS

**Site description and soil collection.** This study was conducted at the Haifeng Nature Reserve in Yunnan Province, China (see Fig. S11A in the supplemental material). The site is a humid habitat with typical subtropical plateau monsoon climate. The annual average precipitation is 1,008 mm. The annual average evaporation is 2,069 mm with a relative humidity of 71%. The annual average temperature is 14.5°C, ranging from the lowest (6°C) in January to the highest (19.7°C) in July. This area is a typical karst landform with many karst tiankeng. The soil is typical Yunnan red clay.

In September 2021, based on previous research in this study area, we selected 3 tiankeng sites (Shenxiantang tiankeng, Bajiaxiantang tiankeng, and Shaojiaxiantang tiankeng). More information about the morphological characteristics of the three tiankengs is shown in Table S5 in the supplemental material. Five sampling areas (each 10 by 10 m) were randomly established inside and outside each tiankeng; three randomly selected quadrats (1 m by 1 m) were established within these sampling areas (Fig. S11B). The plant coverage and species were recorded. After removal of a possible organic layer, we used the five-point sampling method to collected 5 topsoil cores from each quadrat. The soil samples were sieved (2 mm) to remove the plant material and stones and mixed as a composite soil sample for each sampling area. Thus, 10 composite soil samples (subsamples) of each tiankeng were obtained (a total of 30 subsamples). All obtained soil samples were divided into two parts; one part was used for DNA extraction, and the other part were used to determine the soil physicochemical properties. The soil water content (SWC), soil organic C (SOC), total N (TN), total P (TP), available phosphorus (AP), available nitrogen (AN), available potassium (AK), and soil pH values were measured as previously described (67).

**DNA extraction and 16S rRNA gene sequencing.** Total DNA was extracted by cetyltrimethylammonium bromide (CTAB) method (68). DNA concentration and purity were examined with 1% agarose gels. 16S rRNA genes of V4 regions were amplified using the specific primer set 515F (5′-GTGCCAGCMGCCGCGGTAA-3′) and 806R (5′-GGACTACHVGGGTWTCTAAT-3′) with the barcode. All PCRs were done using a Phusion high-fidelity PCR master mix (New England BioLabs) according to the manufacturer's protocol. To minimize amplification bias, three individual PCR amplifications were performed per sample. Sequencing libraries were generated according to standard protocol by TruSeq DNA PCR-free sample preparation kit (Illumina, USA). The library quality was assessed on the Qubit 2.0 fluorometer (Thermo Scientific). The library was sequenced (250-bp paired-end reads) on an Illumina NovaSeq platform.

**Bioinformatics analysis.** We used the customized program scripts of Qiime2docs "Atacama soil microbiome tutorial" (https://docs.qiime2.org/2019.1/) to filter and analyze the raw data (69). Briefly, raw data FASTQ files converts the format via qiime tools, and demultiplexed sequences were quality filtered using QIIME 2 (version 2019.4) for our bioinformatics pipeline. The forward and reverse reads retain a length of 249 bp. The feature table of amplicon sequence variant (ASV) formation was done using the QIIME2 data set 2 plugin and a total of 2,247,020 quality sequences for downstream analyses. The GREENGENES 13_8 database was used for taxonomy classification (70). The rarefaction curves were calculated by Mothur (v.1.43.0) (71). The Chao1 richness and Shannon diversity index were calculated by QIIME2. Based on the beta diversity distance of Bray-Curtis, the profiles of microbial communities inside

and outside of the karst tiankeng were analyzed by principal coordinate analysis (PCoA) (72). The analysis of similarities (ANOSIM), multiresponse permutation procedure (MRPP), and permutational multivariate analyses of variance (PERMANOVA) were used to determine the dissimilarity of microbial communities inside and outside of the tiankeng and conducted using the R package "vegan" (v 4.1.2). The linear discriminant analysis effect size (LEfSe) was applied to reveal the potential microbial taxa biomarkers (http://huttenhower.sph.harvard.edu/galaxy/root?tool_id=PICRUSt_normalize). The STAMP bioinformatic software was used to test microbial taxa and functional gene differences between the inside and outside soil samples ($P < 0.05$) (73). Redundancy analysis (RDA) was performed to reveal the relationship between soil microbial communities and environmental factors using the R package "vegan." The partial Mantel test between microbial communities and environmental factors (9,999 permutations) was computed using the R package "vegan." The functional profiles of microbial communities were predicted by using PICRUSt2 (74).

The assembly processes of microbial communities were determined by the phylogenetic and null model (75). The degree of nonrandom phylogenetic community structure was determined by mean nearest taxon distance (MNTD). Nearest taxon index (NTI) quantifies the size of the normalized effect between MNTD observations and the mean zero distribution. NTI values greater than 0 ($P < 0.05$) indicate that microbial communities are more affected by phylogeny (76). $\beta$-nearest taxon index ($\beta$NTI) and $\beta$-mean nearest taxon distance ($\beta$MNTD) were used to describe phylogenetic $\beta$ diversity of communities (77, 78). $\beta$NTI values greater than 2 and less than $-2$ indicate heterogeneous selection of deterministic processes and homogeneous selection of deterministic processes, respectively; if $\beta$NTI is between $-2$ and 2, it indicates random processes. The MNTD, NTI, $\beta$NTI, and $\beta$MNTD were analyzed by picante package in R software. Raup-Crick matrix (RCbray) can further identify stochastic processes via the vegan package (78).

The construction of the network was based on the principle of molecular ecological network analyses pipeline (79, 80). The network topological properties were calculated in MENA (http://ieg4.rccc.ou.edu/mena/). The ecological role of node was characterized by within-module connectivity ($Zi$) and among-module connectivity ($Pi$). The network hubs ($Zi \geq 2.5$ and $Pi \geq 0.62$), module hubs ($Zi \geq 2.5$ and $Pi \leq 0.62$), and connector network hubs ($Zi < 2.5$ and $Pi \geq 0.62$) were considered as generalists. Visualization of the network was conducted in Cytoscape (v3.9.0) (81). The natural connectivity index was used to reveal the robustness of the microbial communities network (82). Meanwhile, the robustness, vulnerability, and natural connectivity index were used to assess network stability and were analyzed in R software. Detail R code is shown in Yuan et al. (83).

**Data availability.** Raw sequence reads were submitted to the SRA at the NCBI under accession number PRJNA843191.

## SUPPLEMENTAL MATERIAL

Supplemental material is available online only.
**SUPPLEMENTAL FILE 1**, PDF file, 1.1 MB.

## ACKNOWLEDGMENTS

This work was funded by the project "Shenzhen Fundamental Research Program" (GXWD20201231165807007-20200812142216001).

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
