## [Reviewer comments · Microbiology Spectrum]

Microbiology Spectrum

The unique habitat of karst tiankengs changes the taxonomy and potential metabolism of soil microbial communities

Cong Jiang and Hui Zeng

Corresponding Author(s): Hui Zeng, Peking University

Review Timeline:

Submission Date:	June 20, 2022
Editorial Decision:	November 23, 2022
Revision Received:	November 29, 2022
Editorial Decision:	December 3, 2022
Revision Received:	December 4, 2022
Accepted:	December 12, 2022

Editor: Frédérique Reverchon

Reviewer(s): The reviewers have opted to remain anonymous.

Transaction Report:

DOI: <https://doi.org/10.1128/spectrum.02316-22>

November 23, 2022

Prof. Hui Zeng
Peking University
Yiheyuan Road, Haidian District
Beijing, Please select 100871
China

Re: Spectrum02316-22 (The unique habitat of karst tiankengs changes the taxonomy and potential metabolism of soil microbial communities)

Dear Prof. Hui Zeng:

Your manuscript has been revised by two independent reviewers, who agree that this work could be published after some substantial revisions.

Their concerns, which I share, include the lack of details in the Methods section, in particular referring to the sampling design (number of replicates), the choice of some cutoff values and flow of analyses. They also emphasise that most analyses were predictive, which means that your conclusions should be taken with caution. 16S amplicon sequencing is not "metagenomics" and functional inferences are only tentative. Reviewer #1 also highlights the need for more in-depth discussion.

Link Not Available

Sincerely,

Frédérique Reverchon

Journals Department
Reviewer comments:

Reviewer #1 (Comments for the Author):

Overall the results are interesting and the manuscript is well organized and written. However, some main issues not clearly addressed retard me to make fair evaluation of the work. My main concerns are:

1. The overall impression about the research is more description and lack of in-depth discussion about the results.
2. Sampling strategy fundamentally matters the scientific questions you want to address. Much detailed information about sampling should be given. For example, do you have biological triplicates at each sampling plot? Tiankeng usually has dense coverage of plantation. When you do sampling, how to avoid the impact of vegetation on microbial communities?
3. The main conclusions about microbial function in ITK and OTK are not fully discussed, which is not convincing. Why do you see microbial functions related to human diseases in OTK? Are your study areas impacted by human activities? The rationality of the results should be explained based on the situations of your study area.
4. Introduction: unique characteristics of Tiankeng habitats should be clearly stated and also about the study of biota in Tiankeng. Scientific questions and hypothesis based on what you have known should be clearly addressed.
5. Some results and conclusion are contradictory. For example Line 236-238 The results showed that bacteria communities related to human diseases were significantly higher in ITK, which is not consistent with the statement you made in your abstract and conclusion.
6. Some results are obviously not reasonable. Line 232-233: I am curious how can you get the functions related to eukaryotes since you use 16S rRNA sequencing data. Something must be wrong.
7. Line 252 Another example for your statement. Cheng et al., Contrasting bacterial communities and their assembly processes in karst soils under different land use, 2021, Science of the Total Environment, <https://doi.org/10.1016/j.scitotenv.2020.142263>

Other Minor concerns are listed below

- 1 . Line 66: please add the reference for the statement. Or delete the sentence since you will not have anything related to survival strategy later on.
- 2 . Line 71: IN fact mineral substrate significantly impact bacterial and fungal communities in karst caves, please see the ref of Wang et al., The Characterization of Microbiome and Interactions on Weathered Rocks in a Subsurface Karst Cave, Central China. 2022,Front. Microbiol. 13:909494. doi: 10.3389/fmicb.2022.909494
- 3 . Line 73 Please see the ref of Yun et al., (2016) The Relationship between pH and Bacterial Communities in a Single Karst Ecosystem and Its Implication for Soil Acidification. Front. Microbiol. 7:1955. doi: 10.3389/fmicb.2016.01955" for another example of your statement, which clearly see the differences in microbial communities in karst cave and overlying soils.
- 4 . Line76 what do you mean by saying "the survival activities"?
- 5 . Line 145 IT should be ITK
- 6 . Line 226: metagenomics is not the precise word to use here since your study is based on 16S rRNA sequencing.
- 7 . Line 258 the soil in ITK was more nutritious. The statement should be clearly demonstrated by your analysis of physicochemical parameters. You can address this point based on your results rather than the broad speaking.
- 8 . Line 270-272: Something is missing between your discussion and your conclusion sentence in this paragraph. I will strongly recommend you to discuss your results based on the specific unique traits of the Tiankeng.
- 9 . Line 276: Previously you mentioned that ITK was more nutritious, why OTK has more abundant C cycle related genes?
- 10 . Linw 285 : "(32) reported that" , it is not formally used a reference like this.
- 11 . Line 298 what do you mean by saying "greater interactions"?
- 12 . Line 299 more is not used correctly here.
- 13 . Line 301-302 rephrase the sentence
- 14 . Line 308 what are the differences in modules?
- 15 . Line 310-312 what specific habitat do you have for module 1 and module 0 except ITK and OTK? If you do not discern other habitats inside TK, how do you explain different modules within ITK network? You might explain your results in other way since they are in a same habitat ITK or OTK.
- 16 . Line 312-315 What is the logic link between this sentence with the previous one? You mentioned the nodes were mainly positively linked indicating more corporation between microbial groups. Do you see more negative links in OTK than in ITK?
- 17 . Line 328-329 The sentence doesnot make any contribution here.
- 18 . Line ee332-333: rephrase the sentence
- 19 . Line 399: please specify the region you amplified

Reviewer #3 (Comments for the Author):

Summary:

This study focused on the bacterial communities in karst tiankeng ecosystems. Because of their biodiversity and the threat to it due to climate change, understanding the ecosystem's contribution to nutrient cycling via microbial processes is relevant.

Using 16S amplicon sequencing, they identified taxa that were more predominant in the karst tiankeng, compared to outside of it. Shannon diversity calculations confirmed the significance of these differential compositions. Their metadata included physiochemical properties of the two soil types, which correlated to presence of particular taxa. These conclusions were supported by the data and analyses.

There was also a good deal of analysis of the bacterial communities beyond taxonomic classification, most of which relied

predictive programs (PICRUSt, STAMP, and MENA). Although the conclusions based on these predictive programs are valid, the conclusions should be more tempered. Had the analyses been based on whole genome metagenomics or metatranscriptomics, the inferences would hold greater weight.

Are you convinced that the data presented support the main conclusions?

Major points:

1. The figures and tables embedded in the manuscript are very difficult to read. In one case (Table 2), the formatting is affected. They are not included as separate files that might be more readable.
2. Bacterial network analysis informs the main conclusions in this paper. In line 176 it is mentioned that taxa present in >5% of the modules was used to analyze phylogenetic composition of networks. It is not clear why 5% was chosen. It may significantly affect the network analysis, so justification of 5% is called for.
3. As I understand it, 5 soil samples in each of three plots were analyzed for ITK and OTK areas. I would argue these are biological replicates, but I don't see any technical replicates described. Because of the heavy use of predictive programs, technical replicates are all the more important. If technical replicates were included, it needs to be emphasized in the procedures.
4. Since QIIME2 was used, genus level taxonomic identification can be found. The network analyses were done using phyla. It would be helpful if the authors explained why they used the broader taxonomical classification for their analysis. It would be more informative if genus, or at the least family, was used.
5. I find the description of the QIIME2 workflow used a bit vague. Rather than referencing one of the QIIME2 tutorials, it would be more informative if the workflow description provided more detail, including rarefaction curves in the supplemental materials, trimming parameters for both forward and reverse reads, number of filtered sequences, and rarefaction depth.

Minor points:

Lines 64-65: Clarify if community composition is referring to microbial or macroorganism composition.

Line 76: I am unsure what moisturizing is referring to.

Line 80: I would clarify that the study focuses on bacterial communities not all microbes.

Line 87: Denoising was done using DADA2 plugin, so the output is ASVs and not OTUs.

Line 93: What does 13.14 refer to?

Lines 101-112: In line 102, it is said that Actinobacteria is found in both ITK and OTK sites. In lines 109-112, it is said that Actinobacteria were "specific" to ITK. These two statements do not align with one another. If Actinobacteria were found in both sites, how could it then be specific to ITK?

Figure 3A: Does "whole" refer to both sites combined together in the analysis?

Lines 145-146: IT needs to be changed to ITK. I suggest clarifying that the Shannon-Wiener index and species richness, although higher in the ITK, are not significantly different than in in OTK sites.

Table 1: Units for the different physiochemical properties are lacking. It is also unclear what the designations underneath each measurement indicates, but I am assuming it is a site designation. Soil bulk density values are missing in the table.

Line 231: Metabolism of terpenoids and polyketides is the level 2 category.

Line 258: Rather than saying the soil in the tiankeng was more nutritious, it would be more informative if the physiochemical property designation(s) were named.

Line 274: I am unsure what multiple strategies is referring to.

Line 285. The sentence begins with (32) rather than the author's name.

Lines 316, 328: keystone species rather than key species.

Line 326; 332: have participated (past tense); have indicated

Line 380: Is the annual average precipitation for the collection year?

Line 392: Were physiochemical properties determined for each soil sample, or were the samples combined? If so, only one value is given in table 1 for each property, which may be an average.

Lines 393-396: AK is missing from soil tests that were performed.

Line 398: reference for CTAB method is missing. In this same line, there is no indication if the DNA was quantified before amplification.

Line 401: PCR reactions were done...

Line 403: Was any size selection done after amplification? Were the libraries quantified before sequencing?

Line 409: QIIME2 version is not given.

Line 422: STAMP bioinformatic software should be specified, rather than just STAMP.

Line 428: Was PICRUSt or PICRUSt2 used? The latter has more features and bugs that have been fixed.

Line 448: Cytoscape reference is missing.

Line 451: What R package is being referred to here?

Line 457: There is an extra " used.

Line 623: This article is found on pages 581-583.

Line 635: This article is found on pages 814-821.

Staff Comments:

Preparing Revision Guidelines

Please return the manuscript within 60 days; if you cannot complete the modification within this time period, please contact me. If you do not wish to modify the manuscript and prefer to submit it to another journal, please notify me of your decision immediately so that the manuscript may be formally withdrawn from consideration by Microbiology Spectrum.

Overall the results are interesting and the manuscript is well organized and written. However, some main issues not clearly addressed retard me to make fair evaluation of the work. My main concerns are:

1. The overall impression about the research is more description and lack of in-depth discussion about the results.
2. Sampling strategy fundamentally matters the scientific questions you want to address. Much detailed information about sampling should be given. For example, do you have biological triplicates at each sampling plot? Tiankeng usually has dense coverage of plantation. When you do sampling, how to avoid the impact of vegetation on microbial communities?
3. The main conclusions about microbial function in ITK and OTK are not fully discussed, which is not convincing. Why do you see microbial functions related to human diseases in OTK? Are your study areas impacted by human activities? The rationality of the results should be explained based on the situations of your study area.
4. Introduction: unique characteristics of Tiankeng habitats should be clearly stated and also about the study of biota in Tiankeng. Scientific questions and hypothesis based on what you have known should be clearly addressed.
5. Some results and conclusion are contradictory. For example Line 236-238 The results showed that bacteria communities related to human diseases were significantly higher in ITK, which is not consistent with the statement you made in your abstract and conclusion.
6. Some results are obviously not reasonable. Line 232-233: I am curious how can you get the functions related to eukaryotes since you use 16S rRNA sequencing data. Something must be wrong.
7. Line 252 Another example for your statement. Cheng et al., Contrasting bacterial communities and their assembly processes in karst soils under different land use, 2021, *Science of the Total Environment*, <https://doi.org/10.1016/j.scitotenv.2020.142263>

Other Minor concerns are listed below

1. Line 66: please add the reference for the statement. Or delete the sentence since you will not have anything related to survival strategy later on.
2. Line 71: IN fact mineral substrate significantly impact bacterial and fungal communities in karst caves, please see the ref of Wang et al., The Characterization of Microbiome and Interactions on Weathered Rocks in a Subsurface Karst Cave, Central China. 2022, *Front. Microbiol.* 13:909494. doi: 10.3389/fmicb.2022.909494
3. Line 73 Please see the ref of Yun et al., (2016) The Relationship between pH and Bacterial Communities in a Single Karst Ecosystem and Its Implication for Soil Acidification. *Front. Microbiol.* 7:1955. doi: 10.3389/fmicb.2016.01955” for another example of your statement, which clearly see the differences in microbial communities in karst cave and overlying soils.
4. Line 76 what do you mean by saying “the survival activities”?
5. Line 145 IT should be ITK
6. Line 226: metagenomics is not the precise word to use here since your study is based on 16S rRNA sequencing.
7. Line 258 the soil in ITK was more nutritious. The statement should be clearly

demonstrated by your analysis of physicochemical parameters. You can address this point based on your results rather than the broad speaking.

8. Line 270-272: Something is missing between your discussion and your conclusion sentence in this paragraph. I will strongly recommend you to discuss your results based on the specific unique traits of the Tiankeng.
9. Line 276: Previously you mentioned that ITK was more nutritious, why OTK has more abundant C cycle related genes?
10. Line 285 : “(32) reported that” , it is not formally used a reference like this.
11. Line 298 what do you mean by saying “greater interactions”?
12. Line 299 more is not used correctly here.
13. Line 301-302 rephrase the sentence
14. Line 308 what are the differences in modules?
15. Line 310-312 what specific habitat do you have for module 1 and module 0 except ITK and OTK? If you do not discern other habitats inside TK, how do you explain different modules within ITK network? You might explain your results in other way since they are in a same habitat ITK or OTK.
16. Line 312-315 What is the logic link between this sentence with the previous one? You mentioned the nodes were mainly positively linked indicating more corporation between microbial groups. Do you see more negative links in OTK than in ITK?
17. Line 328-329 The sentence doesnot make any contribution here.
18. Line ee332-333: rephrase the sentence
19. Line 399: please specify the region you amplified

Response to Reviewer #1

Dear Reviewer,

Thank you for your letter and for the reviewers comments concerning our manuscript entitled “The unique habitat of karst tiankengs changes the taxonomy and potential metabolism of soil microbial communities” (Spectrum02316-22). Those comments are all valuable and very helpful for revising and improving our paper, as well as the important guiding significance to our research. We have studied the comments carefully and have made corrections, which we hope meet with approval. Revised portions are marked in a revision mode in the paper. The main corrections in the paper and the responses to the reviewer’s comments areas are appended below. To facilitate this discussion, we first retype your comments in italic font and then present our responses to the comments.

Main concerns

1. *The overall impression about the research is more description and lack of in-depth discussion about the results.*

Response: According to the reviewer’s comments, we have added the more depth discussion. It mainly includes a discussion of the bacterial communities composition and function of karst tiankeng. More detail were list in discussion section.

2. *Sampling strategy fundamentally matters the scientific questions you want to address. Much detailed information about sampling should be given. For example, do you have biological triplicates at each sampling plot? Tiankeng usually has dense coverage of plantation. When you do sampling, how to avoid the impact of vegetation on microbial communities?*

Response: According to the reviewer’s comments, we have added more detail information about sampling strategy (Line 449-457). Inside and outside each tiankeng was randomly applied to 5 replicates (each 10 × 10 m) of 3 randomly selected quadrats (1 m × 1 m), respectively. After removal of possible organic layer, we used the five-point sampling method to collected 5 topsoil cores from each quadrats. The soil samples were sieved (2 mm) to remove the plant material and stones, and mixed. Thus, 10 composite soil samples (sub-samples) of each tiankeng were obtained (a total of 30 sub-samples).

3. *The main conclusions about microbial function in ITK and OTK are not fully discussed, which is not convincing. Why do you see microbial functions related to human diseases in OTK? Are your study areas impacted by human activities? The rationality of the results should be explained based on the situations of your study area.*

Response: According to the reviewer’s comments, we have added more discussion about microbial function (Line 325-342). In addition, we also found that gene sequences related to human diseases (such as neurodegenerative disease and infectious bacterial and viral diseases) were more abundant in OTK (Fig. 7). Zhou et al., (2018) showed that alpine grassland degradation increases the risk of bacterial involved in human disease. The karst tiankeng is located in a typical karst degradation area, which is characterized by severe rocky desertification, sparse vegetation, and large disturbance of human activities (e.g., grazing). The ecological environment of OTK is at

great risk, and the loss of biodiversity may have a negative health effect. In contrast, limited by topography, the karst tiankeng is less disturbed by human activities, and the soil and vegetation conditions are better, which creates good conditions for the survival of microorganisms. The higher bacterial communities diversity were observed in ITK (Fig. S3). Sandifer et al. (2015) reported that higher microbial diversity may help to reduce human diseases. Although the relationship between karst ecosystems and human health requires further human disease investigations to determine, this phenomenon is noteworthy.

4. *Introduction: unique characteristics of Tiankeng habitats should be clearly stated and also about the study of biota in Tiankeng. Scientific questions and hypothesis based on what you have known should be clearly addressed.*

Response: According to the reviewer's comments, we have added descriptions of the karst tiankeng unique habitat (Line 53-61). The isolation effect of the vertical cliffs makes the internal habitat of the karst tiankeng independent of the external environment, and a unique primitive microclimate with low temperatures, high humidity and low solar radiation has been formed. Karst tiankeng ecosystem bred rich and unique biological resources. The karst tiankeng flora is characterized by high diversity and strong originality, and preserves ancient and unique plant species such as alder and cool-adapted plants. The karst tiankeng is also a paradise for rare animals, e.g., *Belisana zhangi* sp. nov., *Prionodon pardicolor* and *Sinocyclocheilus hyalinus*.

5. *Some results and conclusion are contradictory. For example Line 236-238 The results showed that bacteria communities related to human diseases were significantly higher in ITK, which is not consistent with the statement you made in your abstract and conclusion.*

Response: Thank you for the suggestion, we have some clerical errors in the original manuscripts, which we have checked and revised in its entirety. In addition, we found that bacteria communities related to human diseases (infectious disease: bacterial and viral) were significantly higher in OTK (Fig. 7).

6. *Some results are obviously not reasonable. Line 232-233: I am curious how can you get the functions related to eukaryotes since you use 16S rRNA sequencing data. Something must be wrong.*

Response: Thank you for the detailed review. By consulting the literature and consulting with the relevant technicians, there may be a misconception here. When the function is predicted, the abundance of orthologous proteins (KEGG ortholog) is predicted by sequence information (this prediction process is not related to species), and orthologous proteins are proteins that have homology in different species. The reason why bacteria predict eukaryotic pathways is because bacteria may have direct homologous proteins with eukaryotic microorganisms, and these proteins are mainly involved in eukaryotic pathways, so eukaryotic pathways will appear in this result. To avoid ambiguity, we have removed this metabolic pathway in Fig. 7.

7. *Line 252 Another example for your statement. Cheng et al., Contrasting bacterial communities and their assembly processes in karst soils under different land use, 2021, Science of the Total Environment, <https://doi.org/10.1016/j.scitotenv.2020.142263>*

Response: According to the reviewer's comments, we have revised the statement according to the

corresponding literature. In our study, diverse taxonomic and functional of bacterial communities were detected in inside and outside karst tiankeng soils with dominance of *Proteobacteria* and *Actinobacteria*, which is consistent with previous study of karst soils (23-26).

Cheng X, Yun Y, Wang H, Ma L, Tian W, Man B, Liu C. 2021. Contrasting bacterial communities and their assembly processes in karst soils under different land use. Science of the Total Environment 751.

Other Minor concerns

1. **Line 66: please add the reference for the statement. Or delete the sentence since you will not have anything related to survival strategy later on.**

Response: Thank you for the detailed review, we have added the reference.

Zhou H, Zhang DG, Jiang ZH, Sun P, Xiao HL, Wu YX, Chen JG. 2019. Changes in the soil microbial communities of alpine steppe at Qinghai-Tibetan Plateau under different degradation levels. Science of the Total Environment 651:2281-2291.

2. **Line 71: IN fact mineral substrate significantly impact bacterial and fungal communities in karst caves, please see the ref of Wang et al., The Characterization of Microbiome and Interactions on Weathered Rocks in a Subsurface Karst Cave, Central China. 2022,Front. Microbiol. 13:909494. doi: 10.3389/fmicb.2022.909494**

Response: Thank you for the detailed review, we have revised it. In fact, mineral substrate significantly impact bacterial and fungal communities in karst caves.

Wang Y, Cheng X, Wang H, Zhou J, Liu X, Tuovinen OH. 2022. The Characterization of Microbiome and Interactions on Weathered Rocks in a Subsurface Karst Cave, Central China. Frontiers in Microbiology 13:909494.

3. **Line 73 Please see the ref of Yun et al., (2016) The Relationship between pH and Bacterial Communities in a Single Karst Ecosystem and Its Implication for Soil Acidification. Front. Microbiol. 7:1955. doi: 10.3389/fmicb.2016.01955" for another example of your statement, which clearly see the differences in microbial communities in karst cave and overlying soils.**

Response: Thank you for the detailed review, we have revised it. Yun et al. (2016) results indicated that compositional variability among microbial communities in the different karst cave habitats reflect spatial pH changes.

Yun Y, Wang H, Man B, Xiang X, Zhou J, Qiu X, Duan Y, Engel AS. 2016. The Relationship between pH and Bacterial Communities in a Single Karst Ecosystem and Its Implication for Soil Acidification. Frontiers in Microbiology 7.

4. **Line76 what do you mean by saying "the survival activities"?**

Response: Thank you for the detailed review. We consider the term “survival activities” is inaccurate and changed to “living activities”. In the isolated karst tiankeng ecosystem, microbial living activities and interactions are essential for maintaining community stability and ecosystem function.

5. **Line 145 IT should be ITK**

Response: Thank you for the detailed review, we have revised it.

6. Line 226: metagenomics is not the precise word to use here since your study is based on 16S rRNA sequencing.

Response: Thank you for the detailed review, we have revised it. The potential functions of karst tiangkeng soil bacterial communities were annotated, and a total of 7342 KO genes were screened.

7. Line 258 the soil in ITK was more nutritious. The statement should be clearly demonstrated by your analysis of physicochemical parameters. You can address this point based on your results rather than the broad speaking.

Response: Thank you for the detailed review, we have revised it. This result is also confirmed by the significantly higher soil nutrient contents of TN, AK, AP and AN in ITK (Table 1).

8. Line 270-272: Something is missing between your discussion and your conclusion sentence in this paragraph. I will strongly recommend you to discuss your results based on the specific unique traits of the Tiangkeng.

Response: Thank you for the detailed review, we have revised it. The indicator taxa in ITK also included *Micromonosporaceae* and *Nitrospira* (Fig. 2C). *Micromonosporaceae* play a key role in enhanced cellulose degrading capability, and possibly favored by rich soil organic C condition (35, 36). *Nitrospira* have been illustrated that sensitive to acid and fertilized soils (37). It should be noted that the karst tiangkeng is characterized by fertile soil and rich vegetation, and it was not surprised to observed these biomarkers in ITK. The biomarkers in OTK included *Phenylobacterium* and *Clostridium*. These biomarkers were related to anthropogenic activities (38, 39). Due to the limitations of topography, karst tiangkeng can be less disturbed by anthropogenic activities.

35.Yeager CM, Gallegos-Graves LV, Dunbar J, Hesse CN, Daligault H, Kuske CR. 2017. Polysaccharide Degradation Capability of Actinomycetales Soil Isolates from a Semiarid Grassland of the Colorado Plateau. *Applied and Environmental Microbiology* 83.

36.Raut S, Polley HW, Fay PA, Kang S. 2018. Bacterial community response to a preindustrial-to-future CO2 gradient is limited and soil specific in Texas Prairie grassland. *Global Change Biology* 24:5815-5827.

37.Le Roux X, Bouskill NJ, Niboyet A, Barthes L, Dijkstra P, Field CB, Hungate BA, Lerondelle C, Pommier T, Tang J, Terada A, Tourna M, Poly F. 2016. Predicting the Responses of Soil Nitrite-Oxidizers to Multi-Factorial Global Change: A Trait-Based Approach. *Frontiers in Microbiology* 7.

38.Farnleitner AH, Ryzinska-Paier G, Reischer GH, Burtscher MM, Knetsch S, Kirschner AKT, Dirnboeck T, Kuschnig G, Mach RL, Sommer R. 2010. *Escherichia coli* and enterococci are sensitive and reliable indicators for human, livestock and wildlife faecal pollution in alpine mountainous water resources. *Journal of Applied Microbiology* 109:1599-1608.

39.Zhong J, Tang H, Li Z, Dong W, Wei C, Li Q, He T. 2021. Effects of combining green manure with chemical fertilizer on the bacterial community structure in karst paddy soil. *Journal of Plant Nutrition and Fertilizer* 27:1746-1756.

9. Line 276: Previously you mentioned that ITK was more nutritious, why OTK has more

abundant C cycle related genes?

Response: Thank you for the detailed review (Line 319-325). The “nutrient limitation theory” can explain these results well. When nutrients are deficient in the soil, increased expression of C and N cycle genes in the microbial community contributes to the breakdown of organic matter, thereby increasing soil nutrients (e.g., C and N). When the soil nutrients are sufficient, the microbial community easily obtains nutrients (Table 1), and the expression of C and N cycle genes will decrease, thus reaching a stable state (38).

Cherif M, Loreau M. 2007. Stoichiometric constraints on resource use, competitive interactions, and elemental cycling in microbial decomposers. American Naturalist 169:709-724.

10. Linw 285 : "(32) reported that" , it is not formally used a reference like this.

Response: Thank you for the detailed review, we have revised it.

11. Line 298 what do you mean by saying "greater interactions"?

Response: Thank you for the detailed review, we consider the term “greater interactions” is inaccurate. The OTK network is more complex than the ITK network, indicating that the interaction of different microbial groups in OTK were more closely.

12. Line 299 more is not used correctly here.

Response: Thank you for the detailed review, we have revised it.

13. Line 301-302 rephrase the sentence

Response: Thank you for the detailed review, we have revised it. Therefore, karst tiankeng microbial communities may be highly sensitive to environmental changes.

14. Line 308 what are the differences in modules?

Response: Thank you for the detailed review, we have revised it. There are differences in the number of nodes and taxa of the main modules in ITK and OTK networks, which implies that habitat heterogeneity plays a role in module formation.

15. Line 310-312 what specific habitat do you have for module 1 and module 0 except ITK and OTK? If you do not discern other habitats inside TK, how do you explain different modules within ITK network? You might explain your results in other way since they are in a same habitat ITK or OTK.

Response: Thank you for the detailed review, we have revised it. The OTK soil had the more stable microbial networks in comparison with ITK soil due to high robustness values (Fig. S7).

16. Line 312-315 What is the logic link between this sentence with the previous one? You mentioned the nodes were mainly positively linked indicating more corporation between microbial groups. Do you see more negative links in OTK than in ITK?

Response: Thank you for the detailed review, we have revised it. The OTK soil had the more stable microbial networks in comparison with ITK soil due to high robustness values (Fig. S8). Several studies have shown that competition within the microbial community is more intense under poor nutritional conditions, and competition between microbes enhances the robustness of

the network (46, 47). This phenomenon was also observed in this study with a higher negative interaction edges in OTK compared with that of ITK soils (Table 2).

17. Line 328-329 The sentence does not make any contribution here.

Response: Thank you for the detailed review, we have revised it. In karst tiankeng, keystones contributed to the community structure and functions because of their diverse taxonomic and complex interactions.

18. Line 332-333: rephrase the sentence

Response: Thank you for the detailed review, we have rephrase the sentence. Different habitats are often accompanied by changes in soil condition, vegetation cover and microclimate.

19. Line 399: please specify the region you amplified.

Response: Thank you for the detailed review, we have revised it. 16S rRNA genes of distinct V4 regions were amplified used specific primer set of 515F (5'-GTGCCAGCMGCCGCGGTAA-3') and 806R (5'-GGACTACHVGGGTWTCTAAT-3') with the barcode.

We would like to take this opportunity to thank you for all your time involved and this great opportunity for us to improve the manuscript. We hope you will find this revised version satisfactory.

Sincerely,
Dr. Hui Zeng

Response to Reviewer #3

Dear Reviewer,

Thank you for your letter and for the reviewers comments concerning our manuscript entitled

“The unique habitat of karst tiankengs changes the taxonomy and potential metabolism of soil microbial communities” (Spectrum02316-22). Those comments are all valuable and very helpful for revising and improving our paper, as well as the important guiding significance to our research. We have studied the comments carefully and have made corrections, which we hope meet with approval. Revised portions are marked in a revision mode in the paper. The main corrections in the paper and the responses to the reviewer’s comments areas are appended below. To facilitate this discussion, we first retype your comments in italic font and then present our responses to the comments.

Major points:

- 1. The figures and tables embedded in the manuscript are very difficult to read. In one case (Table 2), the formatting is affected. They are not included as separate files that might be more readable.*

Response: Thank you for the detailed review, we have revised it, and save figure and tables as separate files.

- 2. Bacterial network analysis informs the main conclusions in this paper. In line 176 it is mentioned that taxa present in >5% of the modules was used to analyze phylogenetic composition of networks. It is not clear why 5% was chosen. It may significantly affect the network analysis, so justification of 5% is called for.**

Response: Thank you for the detailed review. In fact, we used all nodes to build the microbial network, and the specific network characteristics are shown in Table 2. However, in module-based network analysis, in order to more clearly show the main modules and their interaction, we chose a filtering threshold > 5%, which does not affect the network analysis. In module-based network analysis, the main modules are considered to play an important role in the network structure (Cheng et al., 2021). In addition, this threshold has been widely used in previous literature (Yuan et al., 2021).

Cheng X, Yun Y, Wang H, Ma L, Tian W, Man B, Liu C. 2021. Contrasting bacterial communities and their assembly processes in karst soils under different land use. Science of the Total Environment 751.

Yuan MM, Guo X, Wu LW, Zhang Y, Xiao NJ, Ning DL, Shi Z, Zhou XS, Wu LY, Yang YF, Tiedje JM, Zhou JZ. 2021. Climate warming enhances microbial network complexity and stability. Nature Climate Change 11:343-348.

- 3. As I understand it, 5 soil samples in each of three plots were analyzed for ITK and OTK areas. I would argue these are biological replicates, but I don't see any technical replicates described. Because of the heavy use of predictive programs, technical replicates are all the more important. If technical replicates were included, it needs to be emphasized in the procedures.*

Response: Thank you for the detailed review. As far as we know, in ecological research, large-scale sampling mainly emphasizes biological replicates, and our sampling can better represent karst tiankeng habitats. Technical duplication emphasizes repeated detection and analysis of the same sample, which can reduce the analysis variation in the experiment. In the step of 16S gene sequencing, to minimize amplification bias, three individual PCR amplifications were performed per sample. For statistical purposes, biological replicates representing the variability of natural populations are even more important. In addition, we also thank the reviewers for their suggestions, which we haven't focused on before and will be considered in future studies.

4. Since QIIME2 was used, genus level taxonomic identification can be found. The network analyses were done using phyla. It would be helpful if the authors explained why they used the broader taxonomical classification for their analysis. It would be more informative if genus, or at the least family, was used.

Response: Thank you for the detailed review. The network analysis of this study was done using phyla, mainly for two reasons. First, we focused on the differences in microbial communities composition at phyla level (Line 110-115), it can be analyzed with phyla-based microbial network, together. Second, most of the current research is based on phyla level to construct microbial networks. Therefore, the construction of microbial networks based on phyla in this study is helpful for comparison with other studies (Ma et al., 2021; Cheng et al., 2021). In the next research, we will use metagenomic approach to analyze the microbial community composition and structure of karst tiankeng, which is known to annotate into more detailed microbial composition, and we will consider analyzing the microbial network based on family or genus. Many thanks to the reviewer for providing new ideas for data analysis.

*Ma LY, Huang XP, Wang HM, Yun Y, Cheng XY, Liu D, Lu XL, Qiu X. 2021. Microbial Interactions Drive Distinct Taxonomic and Potential Metabolic Responses to Habitats in Karst Cave Ecosystem. Microbiology Spectrum 9.*

*Cheng X, Yun Y, Wang H, Ma L, Tian W, Man B, Liu C. 2021. Contrasting bacterial communities and their assembly processes in karst soils under different land use. Science of the Total Environment 751.*

5. I find the description of the QIIME2 workflow used a bit vague. Rather than referencing one of the QIIME2 tutorials, it would be more informative if the workflow description provided more detail, including rarefaction curves in the supplemental materials, trimming parameters for both forward and reverse reads, number of filtered sequences, and rarefaction depth.

Response: Thank you for the detailed review, we have added more description of the QIIME2 workflow. We also added rarefaction curves in the supplemental materials. Briefly, raw data FASTQ files converts the format via qiime tools, and demultiplexed sequences were quality filtered using QIIME 2 (version 2019.4) for our bioinformatics pipeline. The forward and reverse read retain a length of 249 bp. The feature table of amplicon sequence variant (ASV) formation were done using the QIIME2 data2 plugin (62), and a total of 2,247,020 quality sequences for downstream analyses. The GREENGENES 13_8 database was used for taxonomy classification (63). The rarefaction curves was calculated by Mothur (version v.1.43.0) (64).

*62.Amir A, McDonald D, Navas-Molina JA, Kopylova E, Morton JT, Xu ZZ, Kightley EP,*

Thompson LR, Hyde ER, Gonzalez A, Knight R. 2017. Deblur Rapidly Resolves Single-Nucleotide Community Sequence Patterns. Msystems 2.

63.Bokulich NA, Kaehler BD, Rideout JR, Dillon M, Bolyen E, Knight R, Huttley GA, Caporaso JG. 2018. Optimizing taxonomic classification of marker-gene amplicon sequences with QIIME 2's q2-feature-classifier plugin. Microbiome 6.

64.Amato KR, Yeoman CJ, Kent A, Righini N, Carbonero F, Estrada A, Gaskins HR, Stumpf RM, Yildirim S, Torralba M, Gillis M, Wilson BA, Nelson KE, White BA, Leigh SR. 2013. Habitat degradation impacts black howler monkey (*Alouatta pigra*) gastrointestinal microbiomes. Isme Journal 7:1344-1353.

Minor points

1. Lines 64-65: Clarify if community composition is referring to microbial or macroorganism composition.

Response: Thank you for the detailed review, we have revised it. More heterogeneous habitats generally generate increased variation in microbial community composition and can support higher species alpha diversity.

2. Line 76: I am unsure what moisturizing is referring to.

Response: Thank you for the detailed review, there is an inaccurate wording here, we have delete it. In line 53-56, we re-describe the unique habitat characteristics of karst tiankeng. The isolation effect of the vertical cliffs makes the internal habitat of the karst tiankeng independent of the external environment, and a unique primitive microclimate with low temperatures, high humidity and low solar radiation has been formed.

3. Line 80: I would clarify that the study focuses on bacterial communities not all microbes.

Response: Thank you for the detailed review, we have revised it. Thus, we studied the soil bacterial communities from inside and outside the karst tiankeng.

4. Line 87: Denoising was done using DADA2 plugin, so the output is ASVs and not OTUs.

Response: Thank you for the detailed review, we have revised it.

5. Line 93: What does 13.14 refer to?

Response: Thank you for the detailed review, this is a writing error, we have revised it.

6. Lines 101-112: In line 102, it is said that Actinobacteria is found in both ITK and OTK sites. In lines 109-112, it is said that Actinobacteria were "specific" to ITK. These two statements do not align with one another. If Actinobacteria were found in both sites, how could it then be specific to ITK?

Response: Thank you for the detailed review, "specific" is inaccurate here, we have revised it. More specific, Actinobacteria (phyla), Gammaproteobacteria (class), Actinomycetales (order) and Pseudomonas (genus) were abundant in ITK; Burkholderia (genus), Burkholderiaceae (family) and Bradyrhizobium (genus) were abundant in OTK.

7. Figure 3A: Does "whole" refer to both sites combined together in the analysis?

Response: Thank you for the detailed review. It is true as you said, the "whole" refer to both sites combined together in the analysis. We have added a description in Fig. 3. Whole refer to all karst tiangkeng sites of bacterial community.

8. Lines 145-146: IT needs to be changed to ITK. I suggest clarifying that the Shannon-Wiener index and species richness, although higher in the ITK, are not significantly different than in in OTK sites.

Response: Thank you for the detailed review, we have revised it. The Shannon-Wiener index and species richness were higher at the ITK sites, while the difference with OTK sites was not significant ($P > 0.05$; Fig. S6).

9. Table 1: Units for the different physiochemical properties are lacking. It is also unclear what the designations underneath each measurement indicates, but I am assuming it is a site designation. Soil bulk density values are missing in the table.

Response: Thank you for the detailed review, we have revised it. We do not analyze soil BD in this article, and mention soil BD in the methods is a clerical error.

10. Line 231: Metabolism of terpenoids and polyketides is the level 2 category.

Response: Thank you for the detailed review, we have revised it.

11. Line 258: Rather than saying the soil in the tiangkeng was more nutritious, it would be more informative if the physiochemical property designation(s) were named.

Response: Thank you for the detailed review, we have revised it. This result is also confirmed by the significantly higher soil nutrient contents of TN, AK, AP and AN in ITK (Table 1).

12. Line 274: I am unsure what multiple strategies is referring to.

Response: Thank you for the detailed review. This sentence make no contribution here, we have revised it. Understanding functional group responses to karst tiangkeng unique habitats is critical to predicting karst tiangkeng ecological processes.

13. Line 285. The sentence begins with (32) rather than the author's name.

Response: Thank you for the detailed review, we have revised it. Sandifer et al. (2015) reported that higher microbial diversity may help to reduce human diseases.

14. Lines 316, 328: keystone species rather than key species.

Response: Thank you for the detailed review, we have revised it. Identifying keystone species in a community is important for communities with a high degree of diversity and complexity. The disappearance of keystone species may lead to the fragmentation of networks and modules.

15. Line 326; 332: have participated (past tense); have indicated

Response: Thank you for the detailed review, we have revised it. These microorganisms have participated in the formation of karst soils throughout geological history. Previous studies have indicated that the structure and composition of microbial communities changed as the habitat changed.

16. Line 380: Is the annual average precipitation for the collection year?

Response: “Annual average precipitation” represents the average rainfall per year, not refer the collection year.

17. Line 392: Were physiochemical properties determined for each soil sample, or were the samples combined? If so, only one value is given in table 1 for each property, which may be an average.

Response: Thank you for the detailed review. The physiochemical properties determined for each soil sample. In table 1, each property values are means \pm SE. We have added it in Table 1.

18. Lines 393-396: AK is missing from soil tests that were performed.

Response: Thank you for the detailed review, we have added it.

19. Line 398: reference for CTAB method is missing. In this same line, there is no indication if the DNA was quantified before amplification.

Response: Thank you for the detailed review, we have added it. Total DNA was extracted by CTAB method (60). DNA concentration and purity was examined with 1% agarose gels.
60.Guerra V, Beule L, Lehtsaar E, Liao H-L, Karlovsky P. 2020. Improved Protocol for DNA Extraction from Subsoils Using Phosphate Lysis Buffer. Microorganisms 8.

20. Line 401: PCR reactions were done...

Response: Thank you for the detailed review, we have revised it.

21. Line 403: Was any size selection done after amplification? Were the libraries quantified before sequencing?

Response: Thank you for the detailed review, we have added it. The library quality was assessed on the Qubit@ 2.0 Fluorometer (Thermo Scientific). The library was sequenced (250 bp paired-end reads) on an Illumina NovaSeq platform.

22. Line 409: QIIME2 version is not given.

Response: Thank you for the detailed review, we have added it.

23. Line 422: STAMP bioinformatic software should be specified, rather than just STAMP.

Response: Thank you for the detailed review, we have revised it.

24. Line 428: Was PICRUS_t or PICRUS_t2 used? The latter has more features and bugs that have been fixed.

Response: Thank you for the detailed review. The functional profiles of microbial communities was predicted by using the PICRUS_t2.

25. Line 448: Cytoscape reference is missing.

Response: Thank you for the detailed review, we have added it.

Wu J, Barahona M, Tan YJ, Deng HZ. 2010. Natural Connectivity of Complex Networks. Chinese

Physics Letters 27.

26. Line 451: What R package is being referred to here?

Response: Thank you for the detailed review, we have added it. The detail R code was shown in Yuan et al. (2021).

Yuan MM, Guo X, Wu LW, Zhang Y, Xiao NJ, Ning DL, Shi Z, Zhou XS, Wu LY, Yang YF, Tiedje JM, Zhou JZ. 2021. Climate warming enhances microbial network complexity and stability. Nature Climate Change 11:343-348.

27. Line 457: There is an extra " used.

Response: Thank you for the detailed review, we have revised it.

28. Line 623: This article is found on pages 581-583.

Response: Thank you for the detailed review, we have revised it.

29. Line 635: This article is found on pages 814-821.

Response: Thank you for the detailed review, we have revised it.

We would like to take this opportunity to thank you for all your time involved and this great opportunity for us to improve the manuscript. We hope you will find this revised version satisfactory.

Sincerely,
Dr. Hui Zeng

December 3, 2022

Dr. Hui Zeng
Peking University
Haidian
Beijing
China

Re: Spectrum02316-22R1 (The unique habitat of karst tiankengs changes the taxonomy and potential metabolism of soil microbial communities)

Dear Dr. Hui Zeng:

I have carefully reviewed the revised manuscript and the answers provided by the authors to the reviewers. I still find the discussion to be rather superficial: ecological implications are not fully supported, many references are missing to support the statements, and the putative functions of the microbial communities are not discussed. Please see my detailed comments below. Reviewer 1 also commented that "Scientific questions and hypothesis based on what you have known should be clearly addressed". I did not see the hypothesis and research question clearly stated in the Introduction.

INTRODUCTION

L90: as mentioned by Reviewer 1, "Scientific questions and hypothesis based on what you have known should be clearly addressed".

DISCUSSION

L279-280: "Proteobacteria and Actinobacteria play important roles in phylogenetic, ecological and energy metabolism (14, 27)". Please precise to which concepts "phylogenetic" and "ecological" refer. Do you mean ecological processes, assemblages, phylogenetic selection... It is not clear.

L302-304: "These biomarkers were related to anthropogenic activities (36, 37). Due to the limitations of topography, karst tiankeng can be less disturbed by anthropogenic activities." Please develop, as it is not clear how anthropogenic activities are relevant for OKT. To which anthropogenic activities are you referring to?

L316: "can explain", not "explains". A reference is needed to support this statement.

L316-321: no references are provided regarding these hypotheses.

L329: "the loss of biodiversity may have a negative health effect". Why? Again, there is no reference supporting this statement.

L330-331: what do you mean by "better / good conditions"?

L333: how could microbial diversity help mitigate human diseases? This deserves a more in-depth explanation, I find that the discussion remains rather superficial.

L346-347: what do you mean by "more closely"? This also remains superficial. What are the ecological implications of more complex interactions? Please include references to support your statements.

L365-367: as commented by reviewer 1, spatial heterogeneity within ITK or OTK should also be addressed to discuss the different modules within each habitat. You mentioned heretogeneity in your first version but did not discuss it in detail.

L372-377: the authors state that keystone species are important in complex networks, but emphasize the ecological relevance of keystone species in ITK, which harbours simpler networks than OTK. It seems contradictory.

L384-385: this is a hypothesis, how could you be sure that keystone species participated in community function because of their diversity?

L388-390: references missing to support this statement.

L401: "previous studies", yet only one reference is provided.

L402-405: references are missing.

METHODS

L445: please rephrase. "Five sampling areas (each 10 × 10 m) were randomly established inside and outside each tiankeng; three randomly selected quadrats (1 m × 1 m) were established within these sampling areas.

L450: from each quadrat (remove the "s")

L452: mixed in order to obtain one composite sample per sampling area?

Thank you for submitting your manuscript to Microbiology Spectrum. When submitting the revised version of your paper, please provide (1) point-by-point responses to the issues raised by the reviewers as file type "Response to Reviewers," not in your cover letter, and (2) a PDF file that indicates the changes from the original submission (by highlighting or underlining the changes) as file type "Marked Up Manuscript - For Review Only". Please use this link to submit your revised manuscript - we

strongly recommend that you submit your paper within the next 60 days or reach out to me. Detailed instructions on submitting your revised paper are below.

Link Not Available

Sincerely,

Frédérique Reverchon

Journals Department
Reviewer comments:

Staff Comments:

Preparing Revision Guidelines

Please return the manuscript within 60 days; if you cannot complete the modification within this time period, please contact me. If you do not wish to modify the manuscript and prefer to submit it to another journal, please notify me of your decision immediately so that the manuscript may be formally withdrawn from consideration by Microbiology Spectrum.

Dear Reviewer,

Thank you for your letter and for the reviewers comments concerning our manuscript entitled "The unique habitat of karst tiankengs changes the taxonomy and potential metabolism of soil microbial communities" (Spectrum02316-22). Those comments are all valuable and very helpful for revising and improving our paper, as well as the important guiding significance to our research. We have studied the comments carefully and have made corrections, which we hope meet with approval. Revised portions are marked in a revision mode in the paper. The main corrections in the paper and the responses to the reviewer's comments areas are appended below. To facilitate this discussion, we first retype your comments in italic font and then present our responses to the comments.

1. *Reviewer 1 also commented that "Scientific questions and hypothesis based on what you have known should be clearly addressed". I did not see the hypothesis and research question clearly stated in the Introduction.*

Response: Thank you for the detailed review. We have added the hypothesis in the introduction. Due to the differences in microclimate, soil nutrient, and vegetation cover from inside and outside the karst tiankeng, we hypothesize that (1) karst tiankeng unique habitat alters structure and function of soil microbial communities; (2) karst tiankeng maintain a complex and stable soil microbial network. To test these hypotheses, we studied the soil bacterial communities from inside and outside the karst tiankeng.

2. *L90: as mentioned by Reviewer 1, "Scientific questions and hypothesis based on what you have known should be clearly addressed".*

Response: Thank you for the detailed review. The detailed responses are listed in Q1.

3. L279-280: "Proteobacteria and Actinobacteria play important roles in phylogenetic, ecological and energy metabolism (14, 27)". Please precise to which concepts "phylogenetic" and "ecological" refer. Do you mean ecological processes, assemblages, phylogenetic selection... It is not clear.

Response: Thank you for the detailed review. The statement in the original article is inappropriate and we have deleted and revised it. Proteobacteria are considered to have strong oxidation capacity of organic and inorganic compounds and harvest energy from light (27). The dominant phylum of Proteobacteria in karst tiankeng may suggested that microbes that perform the matter decomposition and nutrient cycles survive well in karst tiankeng.

27 Mukhopadhyaya I, Hansen R, El-Omar EM, Hold GL. 2012. IBD-what role do Proteobacteria play? Nature Reviews Gastroenterology & Hepatology 9:219-230.

4. L302-304: "These biomarkers were related to anthropogenic activities (36, 37). Due to the limitations of topography, karst tiankeng can be less disturbed by anthropogenic activities." Please develop, as it is not clear how anthropogenic activities are relevant for OKT. To which anthropogenic activities are you referring to?

Response: Thank you for the detailed review, we have revised it. These biomarkers were related to anthropogenic activities, such as Clostridium, a sensitive indicator for livestock faecal pollution (36, 37). Due to the limitations of topography, karst tiankeng can be less disturbed by agriculture

and grazing activities.

5. L316: "can explain", not "explains". A reference is needed to support this statement.

Response: Thank you for the detailed review, we have revised it. The "nutrient limitation theory" can explain these results well (38).

38 Cherif M, Loreau M. 2007. Stoichiometric constraints on resource use, competitive interactions, and elemental cycling in microbial decomposers. American Naturalist 169:709-724.

6. L316-321: no references are provided regarding these hypotheses.

Response: Thank you for the detailed review, we have added the references.

39 Zhong Y, Yan W, Wang R, Wang W, Shangguan Z. 2018. Decreased occurrence of carbon cycle functions in microbial communities along with long-term secondary succession. Soil Biology & Biochemistry 123:207-217.

7. L329: "the loss of biodiversity may have a negative health effect". Why? Again, there is no reference supporting this statement.

Response: Thank you for the detailed review, we have revised it. Correlation has been observed between loss in biodiversity with increase in human disease frequency (40, 41).

40 Whitmee S, Haines A, Beyrer C, Boltz F, Capon AG, Dias BFD, Ezeh A, Frumkin H, Gong P, Head P, Horton R, Mace GM, Marten R, Myers SS, Nishtar S, Osofsky SA, Pattanayak SK, Pongsiri MJ, Romanelli C, Soucat A, Vega J, Yach D. 2015. Safeguarding human health in the Anthropocene epoch: report of The Rockefeller Foundation-Lancet Commission on planetary health. Lancet 386:1973-2028.

41 Patil RR, Kumar CS, Bagvandas M. 2017. Biodiversity loss: Public health risk of disease spread and epidemics. Annals of Tropical Medicine and Public Health 10:1432-1438.

8. **L330-331: what do you mean by "better / good conditions"?**

Response: Thank you for the detailed review, we have revised it. In contrast, limited by topography, the karst tiankeng is less disturbed by anthropogenic activities, and the higher soil nutrients and vegetation cover conditions are formed, which creates suitable conditions for the survival of microorganisms.

9. **L333: how could microbial diversity help mitigate human diseases? This deserves a more in-depth explanation, I find that the discussion remains rather superficial.**

Response: Thank you for the detailed review, we have revised it. Sandifer et al. (2015) reported that environmental bacterial profiles are closely associated with human diseases, and exposure to microbial biodiversity help to reduce certain human diseases. Human exposure to high soil microbial diversity increases the chance of collisions between microbiomes and thus avoids the loss of specific microbiome traits (43, 44).

43 Blum WEH, Zechmeister-Boltenstern S, Keiblinger KM. 2019. Does Soil Contribute to the Human Gut Microbiome? Microorganisms 7:287.

44 Mills S, Ross RP. 2021. Colliding and interacting microbiomes and microbial communities - consequences for human health. Environmental Microbiology 23:7341-7354.

10. **L346-347: what do you mean by "more closely"? This also remains superficial. What are**

the ecological implications of more complex interactions? Please include references to support your statements.

Response: Thank you for the detailed review, we have revised it. The OTK network is more complex than the ITK network, indicating that OTK has a greater potential for interactions of different microbial groups. Microbial community resist environmental disturbances by forming strong interspecific interactions (48). The microbial community of OTK adapts to complex environmental changes by forming complex interactions.

48 Wang Y, Ye J, Ju F, Liu L, Boyd JA, Deng Y, Parks DH, Jiang X, Yin X, Woodcroft BJ, Tyson GW, Hugenholtz P, Polz MF, Zhang T. 2021. Successional dynamics and alternative stable states in a saline activated sludge microbial community over 9 years. *Microbiome* 9:199.

11. L365-367: as commented by reviewer 1, spatial heterogeneity within ITK or OTK should also be addressed to discuss the different modules within each habitat. You mentioned heretogeneity in your first version but did not discuss it in detail.

Response: Thank you for the detailed review, we have revised it. Previous studies have interpreted modules as niches with significant implications for ecosystem stable (50, 51). The modularity values were higher in the ITK soil network than in the OTK soil network, which may be linked to stronger niche differentiation in the ITK soil than in the OTK soil. In our study, niche differentiation leads to a decrease in the stability of microbial networks (Table 2). Main modules are closely related to environmental factors, and habitat heterogeneity might affect microbial networks through main modules (52). The differences in the number of nodes and taxa of the main modules in ITK and OTK networks confirmed that habitat heterogeneity plays an important role in module formation.

52 Liu S, Yu H, Yu Y, Huang J, Zhou Z, Zeng J, Chen P, Xiao F, He Z, Yan Q. 2022. Ecological stability of microbial communities in Lake Donghu regulated by keystone taxa. *Ecological Indicators* 136:108695.

In addition, we respect the opinions of reviewers. We also indicated that the core of this manuscript is to compare the differences inside and outside the karst tiankeng, and the construction of the microbial network is also based on all inside or outside site data, so it is less important to discuss spatial heterogeneity within ITK or OTK here. Therefore, we focus on habitat differences inside and outside karst tiankeng.

12. L372-377: the authors state that keystone species are important in complex networks, but emphasize the ecological relevance of keystone species in ITK, which harbours simpler networks than OTK. It seems contradictory.

Response: Thank you for the detailed review. There may be a misunderstanding here. The complex network is not refer specifically to OTK, but also refer ITK. The sentence of “the disappearance of keystone species may lead to the fragmentation of networks and modules” was used incorrectly, and already deleted.

13. L384-385: this is a hypothesis, how could you be sure that keystone species participated in community function because of their diversity?

Response: Thank you for the detailed review, we have deleted and revised it. Different keystones play an important role in maintaining differentiated community structure and function inside and

outside karst tiankeng.

14. L388-390: references missing to support this statement.

Response: Thank you for the detailed review, we have added it.

62 Anamulai S, Sanusi R, Zubaid A, Lechner AM, Ashton-Butt A, Azhar B. 2019. Land use conversion from peat swamp forest to oil palm agriculture greatly modifies microclimate and soil conditions. Peerj 7:7656.

15. L401: "previous studies", yet only one reference is provided.

Response: Thank you for the detailed review, we have revised it. Liu et al. (2012) studies have reported that TP is a limiting factor of microbial communities.

Liu L, Gundersen P, Zhang T, Mo JM. 2012. Effects of phosphorus addition on soil microbial biomass and community composition in three forest types in tropical China. Soil Biology & Biochemistry 44:31-38.

16. L402-405: references are missing.

Response: Thank you for the detailed review, we have added it. Karst areas are generally subject to phosphorus restrictions and thus TP content affects the soil microbial communities (65). The soil AK content mainly comes from the accumulation of effective nutrients (66).

65 Chen H, Li D, Xiao K, Wang K. 2018. Soil microbial processes and resource limitation in karst and non-karst forests. Functional Ecology 32:1400-1409.

66 Shen Y, Yu Y, Lucas-Borja ME, Chen F, Chen Q, Tang Y. 2020. Change of soil K, N and P following forest restoration in rock outcrop rich karst area. Catena 186:104395.

17. L445: please rephrase. "Five sampling areas (each 10 × 10 m) were randomly established inside and outside each tiankeng; three randomly selected quadrats (1 m × 1 m) were established within these sampling areas.

Response: Thank you for the detailed review, we have revised it.

18. L450: from each quadrat (remove the "s")

Response: Thank you for the detailed review, we have revised it.

19. L452: mixed in order to obtain one composite sample per sampling area?

Response: Thank you for the detailed review, we have revised it. The soil samples were sieved (2 mm) to remove the plant material and stones, and mixed as a composite soil sample for each sampling area.

We would like to take this opportunity to thank you for all your time involved and this great opportunity for us to improve the manuscript. We hope you will find this revised version satisfactory.

Sincerely,
Dr. Hui Zeng

December 12, 2022

Dr. Hui Zeng
Peking University
Haidian
Beijing
China

Re: Spectrum02316-22R2 (The unique habitat of karst tiankengs changes the taxonomy and potential metabolism of soil microbial communities)

Dear Dr. Hui Zeng:

I am pleased to inform you that your manuscript has been accepted for publication in Microbiology Spectrum. Please consider integrating the following minor edits once receiving the proofs for your article.

L16: predicted functions

L18: most likely due to... (a direct causal relationship was not demonstrated)

L21: remove coma after "network"

L22: keystone species

L27-28: "that some human pathogenic microorganisms" (instead of diseases)

L84: "soil nutrients" (plural). Please include a reference to justify the abiotic differences between inside and outside the karst tiankeng.

L267: remove "of"

L272: Please consider changing reference no. 27, which refers to pathogenic Proteobacteria in the intestinal tract (not relevant here). Also, not all Proteobacteria are photosynthetic.

L317: please rephrase. "increase the risk of potentially pathogenic microorganisms involved in human disease. , and the indiscriminate introduction of livestock in this area has caused Salmonella infections".

L326: please reformulate. "and is characterized by higher soil nutrient contents and larger vegetation cover, which..."

L346: may adapt (this is hypothetical)

L356: ecosystem stability

L359: "niche differentiation led..."

L382: keystone species ... may play (also hypothetical)

Your manuscript has been accepted, and I am forwarding it to the ASM Journals Department for publication. You will be notified when your proofs are ready to be viewed.

Sincerely,

Frédérique Reverchon
